# Solvent-dripping modulated 3D/2D heterostructures for high-performance perovskite solar cells

Xiaoming Chang [1,9], Randi Azmi [2,9], Tinghuan Yang[3], Nan Wu[3], Sang Young Jeong[4], Herui Xi[5], Drajad Satrio Utomo[1], Badri Vishal[1], Furkan H. Isikgor [1], Hendrik Faber [1], Zhaoheng Ling[1], Mingjie He [1], Marco Marengo [1], Pia Dally[1], Adi Prasetio [1], Yu-Ying Yang[1], Chuanxiao Xiao [5,6], Han Young Woo [4], Kui Zhao[3], Martin Heeney[1], Stefaan De Wolf [1], Leonidas Tsetseris [7] & Thomas D. Anthopoulos [1,8] ✉

The controlled growth of two-dimensional (2D) perovskite atop three-dimensional (3D) perovskite films reduces interfacial recombination and impedes ion migration, thus improving the performance and stability of perovskite solar cells (PSCs). Unfortunately, the random orientation of the spontaneously formed 2D phase atop the pre-deposited 3D perovskite film can deteriorate charge extraction owing to energetic disorder, limiting the maximum attainable efficiency and long-term stability of the PSCs. Here, we introduce a *meta*-amidinopyridine ligand and the solvent post-dripping step to generate a highly ordered 2D perovskite phase on the surface of a 3D perovskite film. The reconstructed 2D/3D perovskite interface exhibits reduced energetic disorder and yields cells with improved performance compared with control 2D/3D samples. PSCs fabricated with the *meta*-amidinopyridine-induced phase-pure 2D perovskite passivation show a maximum power conversion efficiency of 26.05% (a certified value of 25.44%). Under damp heat and outdoor tests, the encapsulated PSCs maintain 82% and 75% of their initial PCE after 1000 h and 840 h, respectively, demonstrating improved practical durability.

Organic-inorganic metal halide perovskite solar cells (PSCs) have a verified power conversion efficiency (PCE) above 26%, making them a viable photovoltaic technology[1–3]. However, in terms of operational stability, the commercialization of PSCs faces considerable challenges. Three-dimensional (3D) perovskites degrade mostly due to the high density of defects and ion migration at grain boundaries and interfaces, occurring under practical operating conditions[4–10]. Coating two-dimensional (2D) perovskite layers on the surface of 3D perovskite films has successfully been used to eliminate surface defects and block ion migration, improving the stability and PCE of PSCs[6,7,11–15]. However,

[1]KAUST Solar Center (KSC), Physical and Engineering Division (PSE), King Abdullah University of Science and Technology (KAUST), Thuwal, Kingdom of Saudi Arabia. [2]School of Science and Engineering, The Chinese University of Hong Kong, Shenzhen, Guangdong, China. [3]School of Materials Science and Engineering, Shaanxi Normal University, Xi'an, China. [4]Department of Chemistry, Korea University, Seoul, Republic of Korea. [5]Ningbo Institute of Materials Technology and Engineering, Chinese Academy of Sciences, Ningbo City, China. [6]Ningbo New Materials Testing and Evaluation Center Co. Ltd, Ningbo City, China. [7]Department of Physics, School of Applied Mathematical and Physical Sciences, National Technical University of Athens, Athens, Greece. [8]Henry Royce Institute, Photon Science Institute, Department of Electrical and Electronic Engineering, The University of Manchester, Manchester, UK. [9]These authors contributed equally: Xiaoming Chang, Randi Azmi. ✉e-mail: thomas.anthopoulos@manchester.ac.uk

this method inevitably results in a mixture of various $n$ values of 2D perovskite layers (where $n$ represents the number of inorganic slabs) with random orientation, potentially slowing down charge transfer and reducing the ferroelectric properties (or internal electric field capability) of these layers[11–13]. In addition, the disordered structure of 2D layers can affect the structural instability of 3D/2D heterostructure-based PSCs[11–13]. On the other hand, the persistent challenge of acquiring a phase-pure 2D perovskite layer with ordered structures on 3D perovskite surfaces stems from a limited number of studies exploring the structure of 2D ligands and post-fabrication engineering[6,7,11–14]. Previous studies have so far focused on the traditional fabrication method and ammonium-based ligands[1,2,6,7,16].

Here, we utilized the *meta*-amidinopyridine (MAP) ligand to create a phase-pure 2D perovskite ($n = 1$) region on the surface of a 3D perovskite layer. The MAP ligand has an asymmetric structure with a strong dipole moment, and upon interacting with the perovskite layer, it can reorient the 2D domains from a random orientation to a highly ordered one during the post-dripping steps, significantly improving their ferroelectric properties and passivation effects at the 3D/2D interface without deteriorating charge transport. This led to the development of high-efficiency inverted PSCs with a maximum PCE of 26.05% (certified as 25.44%), making them one of the most efficient PSCs based on 3D/2D heterostructures. The operational stability of the 3D/2D heterostructure PSCs was also improved under damp heat and outdoor tests, with 82% and 75% retention of their initial PCE after 1000 h and 840 h, respectively.

## Results

### Processing of 3D/2D perovskite films

3D/2D perovskite heterostructures are formed by dissolving suitable ligands in polar solvents, such as isopropanol (IPA), and spin-coating the ensuing solution onto the pre-deposited 3D perovskite layer, followed by thermal annealing at 80 °C or higher temperatures. Unfortunately, this method often results in an excess amount of unreacted 2D ligands on the 3D perovskite surface, adversely affecting the 2D perovskite orientation, charge transport, electric field distribution at the interface, and the overall cell stability[7,17–20]. Therefore, an additional post-dripping step is often necessary to remove unreacted excess ligands after 2D perovskite formation[7,19,20]. However, the impact of the post-dripping step on modifying or reconstructing the 2D layer remains largely unknown, particularly when using polar solvents that can partially dissolve organic cations. In addition, the potential for 2D perovskite layer reconstruction depends on the reactivity of the ligands, regardless of whether it removes or refines 2D perovskite layers.

To elucidate the impact of solvent post-dripping with the ligands, we compared MAP with the conventional phenethylammonium (PEA). Grazing-incidence wide-angle X-ray scattering (GIWAXS) analysis (Fig. 1a, b, Figs. S1 and S2) was used to investigate the impact of the different ligands on the 3D/2D-phase formation. The as-deposited perovskite layer without 2D treatment is labeled as the 'control' sample, while the 3D/2D perovskite layers formed without post-dripping are labeled as "2D-PEA-disordered" and "2D-MAP-disordered". Samples subjected to the post-dripping step with the different ligands are labeled "removed-2D-PEA" and "2D-MAP-ordered". The control perovskite sample shows a distinct diffraction peak at $q_z$ (-1 Å$^{-1}$), representing the (100) main peak of 3D perovskite (Fig. S2), with a considerable PbI$_2$ peak at $q_z$ (-0.9 Å$^{-1}$)[7]. This excess PbI$_2$ on the 3D perovskite surface is expected to aid in the 2D perovskite conversion[7,14]. After 2D treatment, this PbI$_2$ peak has disappeared, along with new additional peaks at lower $q_z$ (<0.6 Å$^{-1}$), which is assigned to the 2D perovskite phase. It is noted that 2D-PEA has two 2D perovskite peaks at $q_z = 0.35$ Å$^{-1}$ and 0.27 Å$^{-1}$ (Fig. S2), representing $n = 1$ and 2 phases, respectively. This is consistent with previous works showing 2D-PEA has a mixture of phases[20,21]. Upon applying the first IPA dripping step, all 2D-associated peaks disappear, and the PbI$_2$ peak increases. We hypothesized that, following the solvent post-dripping,

the PEA ligand was dissolved and removed by leaving residual PbI$_2$ on the perovskite surface.

In contrast, 2D-MAP without post-dripping step has only a single 2D perovskite peak at $q_z = 0.49$ Å$^{-1}$ with a full half-ring, indicating single-phase 2D perovskite ($n = 1$) formation with random crystalline domain orientation (Fig. 1a)[22]. Interestingly, following multiple steps of IPA dripping (3rd and 4th), 2D-MAP becomes more oriented, as indicated by the strong peak in the z-direction (parallel orientation). In addition, 2D-MAP maintained its phase purity throughout this process. Applying additional post-dripping steps (up to 10) led to the weakening of the 2D-MAP-ordered peak and enhancement of the PbI$_2$ peak. This analysis suggests that MAP ligands are reacting less with the post-dripping solvent or have stronger binding to the inorganic octahedral perovskite matrices, since 2D-MAP remains present even after several post-dripping/washing steps. Post-dripping also causes a shifting in the $q_z$ peak of the 2D-MAP-ordered film indicating structural reconstruction of the 2D-MAP phase, potentially due to the formation of smaller 2D crystals.

To confirm the effectiveness of IPA post-dripping for removing unreacted 2D ligands, we prepared 3D/2D-MAP films before and after IPA post-dripping. Prior to IPA post-dripping, the 3D/2D films appeared greenish, which faded after four IPA post-dripping cycles and became darker in color (Fig. S3a). Interestingly, after 10 cycles, the color of the 3D/2D-MAP film became similar to that of the control film. Thus, we speculated that the greenish color could be associated with the excess unreacted MAP ligand on the surface of the film. To confirm the existence of 2D-MAP and their proportion, we integrated the 2D peaks from the GIWAXS data for each sample (Fig. S3b). We observed that as the number of post-dripping cycles increased from 0 to 4, the amount of 2D perovskite continuously increased, suggesting that the unreacted MAP ligand was redissolved in IPA and facilitated the reconversion to 2D perovskite with preferable parallel orientation. Subsequently, we also evaluated the impact of IPA post-dripping on 3D perovskite films' quality by analyzing the steady-state PL spectra of untreated 3D films and those treated with PEA and MAP ligands (see Fig. S4). The control 3D film exhibited diminished PL intensity owing to IPA-induced dissolution and defect formation, consistent with previous findings[19]. In contrast, the 2D-passivated films demonstrated increased PL intensity. The 2D-PEA sample displayed a slight PL enhancement post-dripping, indicating improved molecular PEA passivation compared to 2D passivation[23]. The 2D-MAP passivation sample exhibited significant PL enhancement after post-dripping, suggesting improved passivation from the ordered 2D-MAP layer. These results confirm that IPA post-dripping enhances 2D passivation without significantly compromising the 3D/2D-MAP film.

To understand the microstructural evolution caused by the solvent post-dripping step, we performed scanning electron microscopy (SEM) measurements on the perovskite films. Figure S5a–f shows the SEM images of the control and 3D/2D perovskite films, with and without solvent post-dripping. Several changes in the grain size and surface morphology of the ordered 3D/2D-MAP samples compared to the control perovskite sample, occur. Notably, we observed the formation of flake-like structures in the disordered 3D/2D-MAP films, which corresponded to the irregular size of the 2D perovskite layers. Following post-dripping treatment (3D/2D-MAP-ordered films), the flakes disappeared with a more distinct morphology, suggesting a reorganization of the 2D perovskite phase on the surface and potentially within the 3D perovskite layer. Interestingly, the disordered 3D/2D-PEA films exhibit significant changes in the surface morphology with smaller grain sizes, possibly indicating the formation of a 2D-PEA perovskite layer. However, after the post-dripping step, the morphology returns to that of the control perovskite sample accompanied by the presence of large PbI$_2$-rich domains (bright white). The higher solubility of the PEA ligand in IPA compared to that of the MAP ligand appears to be key in facilitating surface reconstruction (see Fig. S5g),

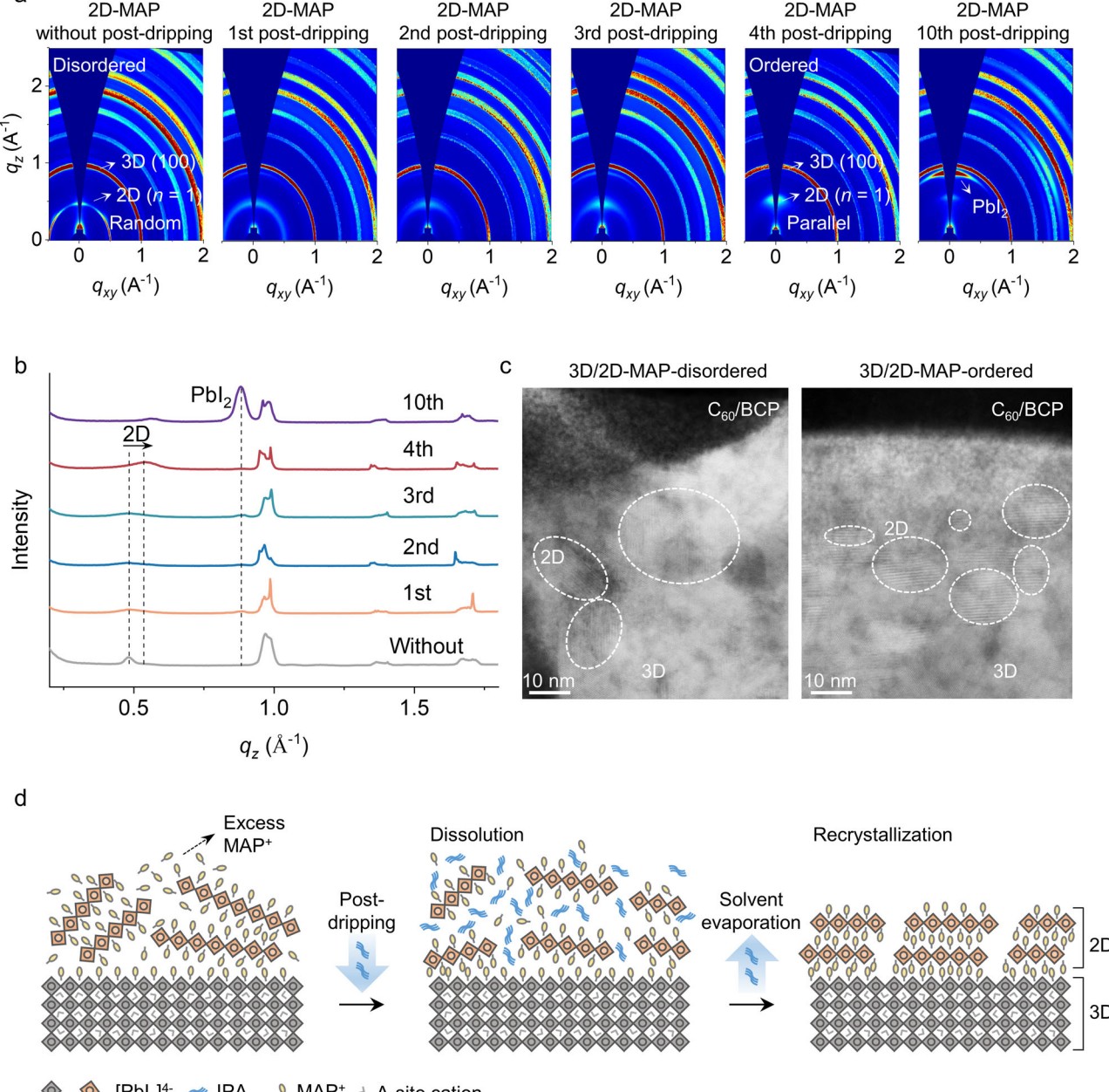

**Fig. 1 | Processing and characterization of 3D/2D perovskite films. a** 2D GIWAXS images of perovskite films with 0.1° X-ray incidence angles. **b** Combined intensity from GIWAXS data in the $q_z$ direction with 0.1° X-ray incidence angles. **c** Cross-sectional HR-STEM images of top 3D/2D interface of the perovskite/$C_{60}$/BCP interfaces. **d** Schematic illustrating the reorientation of the 2D-MAP perovskite during and after the post-dripping process.

involving the removal of the 2D layer and formation of residual $PbI_2$. It should be noted that in the latter sample, the PEA ligand might partially exist on the surface of the bulk 3D perovskite. These findings are consistent with the GIWAXS analysis.

Next, we studied the 2D perovskite layer orientation in the 2D-MAP sample before and after post-dripping steps using cross-sectional high-resolution scanning transmission electron microscopy (HR-STEM, Fig. 1c, Fig. S6). 2D-MAP without post-dripping treatment exhibits a disordered orientation of the 2D phase atop the 3D per-ovskite. In contrast, after solvent post-dripping, the 2D-MAP sample shows a more ordered 2D phase parallel to the 3D perovskite layer with a more infiltrated structure.

We then investigated the chemical binding and interactions between the PEA and MAP ligands on the 3D perovskite surface to further confirm our hypothesis. Figure S7a–c shows an X-ray photoelectron spectroscopy (XPS) analysis of the perovskite films. After MAP treatment, the N 1s peak of the $FA^+$ and $MA^+$ cations shifts to a lower binding energy (0.65 eV) because of the electron-donating ability of the pyridinic nitrogen atoms of MAP. In contrast, the N 1s peak shifts to a higher binding energy (0.24 eV) after PEA treatment, which can be attributed to the electron-withdrawing ability of the $PEA^+$ cations. Additionally, there is no significant shift in the Pb 4f and I 3d peaks, likely due to the weaker interaction between the $PEA^+$ cations and Pb or I atoms. Interestingly, the Pb 4f and I 3d peaks for the MAP-treated sample shift to lower energies of 0.60 and 0.69 eV, respec-tively. The shift can be attributed to the strong interactions between surface Pb atoms and the pyridinic nitrogen atoms on MAP[24]. Mean-while, the shift in the I 3d peaks is expected to be due to the electro-static interaction between the iodide ion (I⁻) and the amidino group in MAP, which can form through hydrogen or ionic bonds[25].

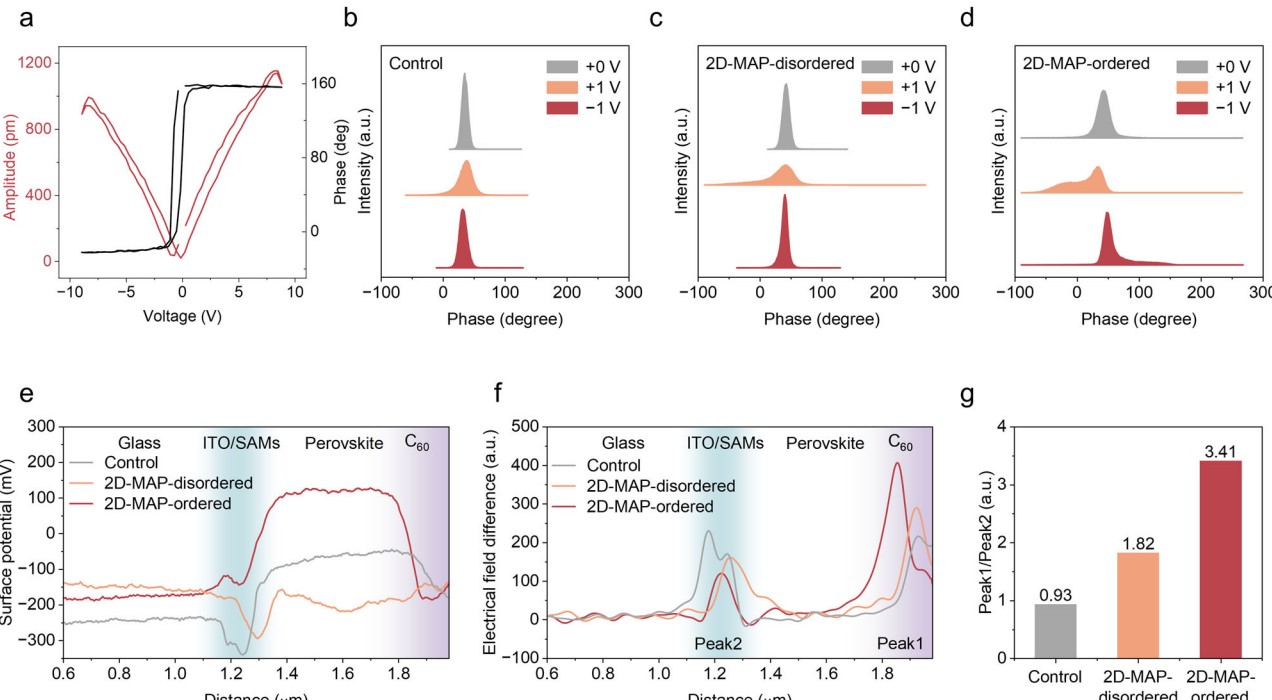

**Fig. 2 | Electric field across the 3D/2D heterostructure. a** Hysteretic dependence of the PFM phase and amplitude with the applied DC bias for 2D-MAP-ordered-based perovskite films. **b–d** PFM phase distribution of the control, 2D-MAP-disordered, and 2D-MAP-ordered-based perovskite films under different bias voltages. **e** Surface potential depth profiles (0 V) and **f** corresponding electric field distribution (under a reverse bias of −1 V) of the control, 2D-MAP-disordered and 2D-MAP-ordered-based devices. **g** Peak ratios of the electric field at the perovskite/C$_{60}$ (Peak1) and SAMs/perovskite (Peak2) interfaces.

Using $^1$H nuclear magnetic resonance (NMR) spectroscopy, we further examined the interactions between MAP$^+$ and PbI$_2$. Upon addition of PbI$_2$, the $^1$H NMR spectra (H proton a of Fig. S7d) revealed a significant downfield shift of the amidino proton resonance of MAP$^+$ cations from δ = 9.14 ppm, suggesting robust hydrogen bonding interactions between MAP$^+$ and I$^-$ [2]. Shifts in the protons of the −CH− group (H proton b, c of Fig. S7d) in the pyridine ring further indicated coordination between Pb$^{2+}$ and the nitrogen of the pyridine ring in MAP$^+$. We also compared our experimental results with those from Density Functional Theory (DFT) calculations. Based on these calculations, it is found that the MAP$^+$ cations have multiple sites of binding to the perovskite surface (Fig. S8) due to their pyridine and amidino groups. For instance, MAP$^+$ cations can interact with both Pb atoms and I$^-$ anions, which can facilitate the formation of phase-pure 2D perovskite domains, which can, in turn, limit ligand penetration deeper into the layer required for the formation of the low $n=1$ phase [26]. Additionally, the multiple sites of binding of MAP$^+$ cation could potentially help to enhance the chemical and structural stability of the 3D/2D heterostructure. In contrast, PEA$^+$ cations interact less strongly with the perovskite surface, so they cannot facilitate the modification of the latter to a stable capping layer as easily.

## Electric field across the 3D/2D heterostructure

We further assessed the effects of ordering 2D passivation structures on the surface potential distribution using Kelvin probe force microscopy (KPFM). The ordered 3D/2D-MAP samples showed reduced surface roughness compared with the 3D/2D-PEA and control samples, as shown in Fig. S9a. The 2D passivation treatment reduces the perovskite films' work function (WF) (Fig. S9b–d). Notably, the MAP molecule has a higher dipole moment (4.06 D) than the PEA molecule (1.91 D), as shown by the electrostatic potentials and dipole moments in Fig. S10. The higher dipole moment of the 2D ligand might strongly induce the WF shift of the 3D/2D perovskite films [27]. However, the 3D/2D-MAP samples without post-dripping exhibit a marginal WF shift,

likely due to the disordered structure of the 2D layer and the presence of unreacted excess ligands on the 3D perovskite surface. In contrast, the ordered 3D/2D-MAP samples exhibit the largest WF shift, which can be attributed to the synergetic effect of the well-ordered 2D layers (Fig. S9c, d). A higher WF shift enhances the band bending (i.e. higher internal electric field) at the 3D/2D interface, which is expected to aid electron extraction.

The higher dipole moment of the ligands and improved ordering of the 2D phase can also affect the ferroelectric properties of the perovskite [28] and hence the internal built-in potential which is crucial for charge separation. Indeed, analysis of the layers using piezo-response force microscopy (PFM) reveals clear ferroelectric behavior in the 3D/2D films (Fig. 2a, Fig. S11). Both ordered and disordered 3D/2D samples exhibit a 180° phase flip with hysteresis at the positive and negative bias voltages, while the control-based perovskite films do not exhibit a noticeable ferroelectric response. By applying different voltages (0, +1, and −1 V), we see a clear distinction in the PFM phase distribution of each sample. The control perovskite film has negligible phase distribution changes under various applied voltage biases, except for the intensity difference, suggesting the absence of ferroelectricity, as shown in Figs. 2b and S12. The disordered 3D/2D-MAP sample shows weak ferroelectricity, as shown in the PFM distribution plot and mapping images (Fig. 2c, Fig. S13). In contrast, the well-ordered 3D/2D-MAP sample displayed a stronger phase angle difference with a highly distinct phase, indicating improved ferroelectric properties (Fig. 2d, Fig. S14).

We also visualized the electric field distribution of the control and 3D/2D devices using cross-sectional KPFM through both interfaces, involving C$_{60}$ electron-transport layers and self-assembled monolayer (SAM) hole-transport layers (see Figs. S15–18). The ordered 3D/2D-MAP-based devices show the largest potential drops and electric field enhancement at the perovskite/C$_{60}$ contact (Fig. 2e, f), benefiting from the enhanced ferroelectricity of the ordered 2D-MAP layer. In addition, the electric field ratio between the perovskite/C$_{60}$ and ITO/SAMs/

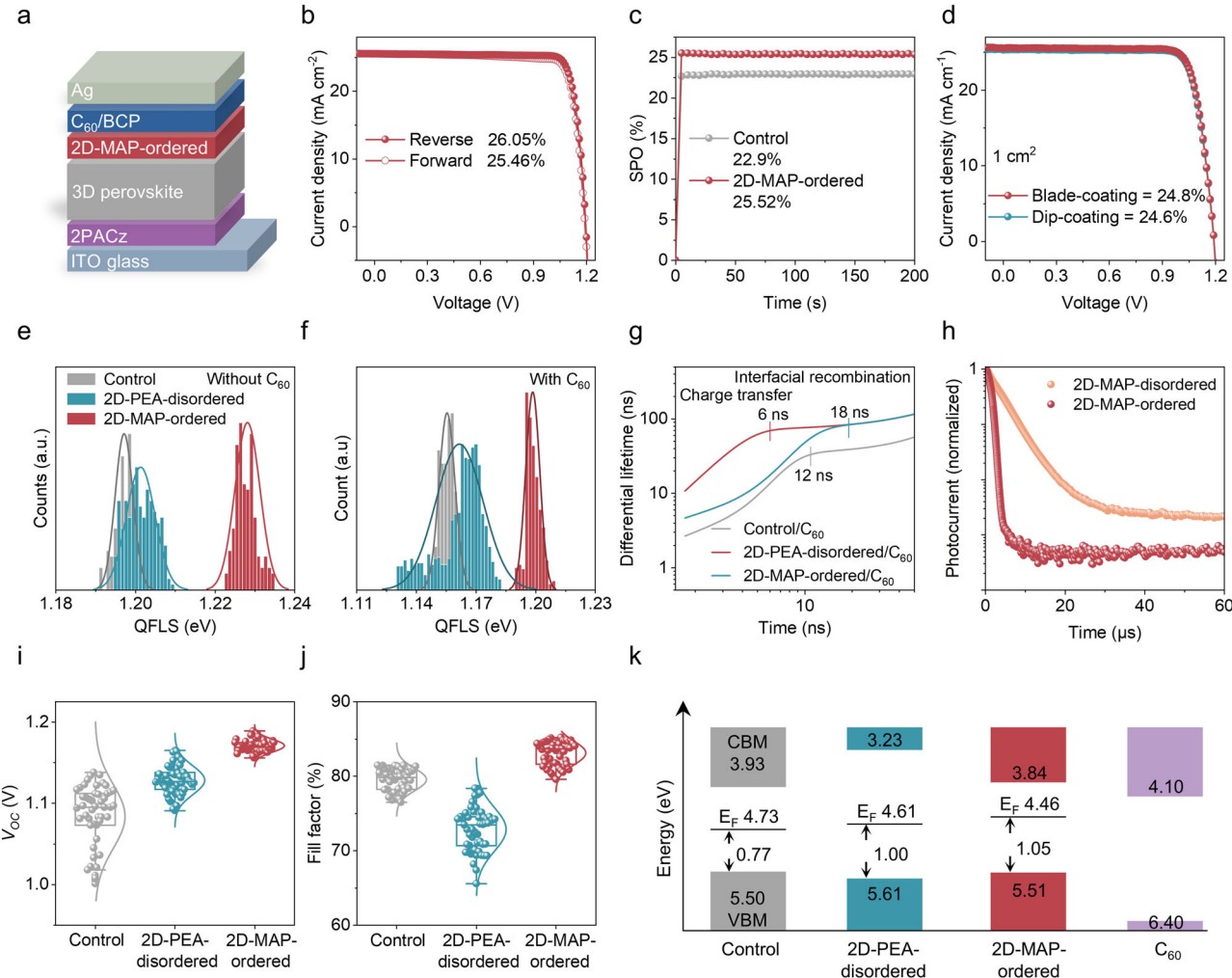

**Fig. 3 | Photovoltaic performance and charge carrier dynamics. a** Schematic structure of inverted perovskite solar cells. **b** $J$–$V$ curves of MAP-treated champion devices (0.063 cm²). **c** SPO of PCE of the control and 2D-MAP-ordered champion cell. **d** $J$–$V$ curves of the 2D-MAP-ordered champion devices with an active area of 1.0 cm². **e**, **f** QFLS distribution of the perovskite films without (**e**) and with $C_{60}$ (**f**).

**g** Differential carrier lifetimes extracted from the time-resolved PL spectra. **h** TPC measurements of 2D-MAP-disordered and 2D-MAP-ordered-based devices. **i**, **j** Statistical comparison of photovoltaic parameters ($V_{OC}$ and FF). **k** Energy level scheme for perovskite films extracted from UPS and LE-IPES data.

perovskite interfaces increased in devices as follows: Control < disordered 3D/2D-MAP < ordered 3D/2D-MAP, indicating the enhancement of 3D/2D junction quality and reduced interfacial defects upon inserting the ordered 2D-MAP layer (Fig. 2g)[29].

## Photovoltaic performance and charge carrier dynamics

We explored the effects of ordered and phase-pure 2D-MAP perovskite passivation on device characteristics by employing the inverted p-i-n cell configuration, Glass/ITO/2PACz/3D perovskite/2D/$C_{60}$/BCP/silver (Fig. 3a). Cells featuring the ordered 2D-MAP passivation layer yielded a champion PCE of up to 26.05% (reverse scan)/25.46% (forward scan), while the control device had a PCE of 23.5% under reverse scan, benefiting from enhancements in all device parameters (Fig. 3b, Table S1). The stabilized power outputs (SPO) of the control and MAP-based devices were 22.9% and 25.5%, respectively (Fig. 3c). We also certified our champion device and obtained a PCE of 25.44% (open-circuit voltage ($V_{OC}$) of 1.196 V, short-circuit current density ($J_{SC}$) of 25.18 mA cm⁻², and fill factor (FF) of 84.48%) by an independent laboratory (Fig. S19). The higher $J_{SC}$ (25.53 mA cm⁻²) measured for the champion cell featuring a perovskite with a bandgap of ≈1.53 eV, is consistent with the value obtained from external quantum efficiency (EQE) measurements (≈1.1% mismatch) (Fig. S20a). The bandgap of the

3D perovskite was determined from the first derivative of the EQE (Fig. S20b). Large-area cells (1.0 cm²) were also fabricated to assess the scalability of our method (Figs. S21, 22). Figure 3d shows the data for a 1.0 cm² active-area cell yielding a PCE of 24.8% and 24.6% through blade-coating and dip-coating of IPA as replacement steps to the post-dripping via spin-coating. These results confirm the compatibility of our method with scalable deposition processes.

Cells featuring the ordered 2D-MAP passivation exhibit a higher $V_{OC}$ by approximately 56 mV, which can be attributed to reduced trap density following the formation of the 2D passivation layer. To quantify the impact of the different surface treatments on the trap concentration in the formed perovskite layers, we measured the quasi-fermi level splitting (QFLS) for the control, 2D-PEA, and 2D-MAP samples. Figure S23a shows the QFLS maps obtained via the hyper-spectral imaging in "half stack" devices composed of Glass/ITO/2PACz/ 3D perovskite/2D with and without the $C_{60}$ layer. The measurements revealed a marked enhancement in the QFLS for MAP-treated samples from 1.197 to 1.227 eV (without $C_{60}$) and 1.155 to 1.193 eV (with $C_{60}$). We further analyzed the $V_{OC}$-loss of each sample and found it to be consistent with the QFLS trend (see Fig. S23b). These drastic changes are attributed to the effective passivation of the perovskite surface upon post-dripping treatment with MAP due to multiple binding sites

(Fig. 3e, f). Notably, the FF of the champion device exceeds 85.5%, approaching 94.4% of the theoretical limit value at the given perovskite bandgap, whereas the FF of the control device remains at 81.45%. This remarkable FF value is most likely attributed to the improved charge carrier extraction and internal electric field at the electron contact junction.

We performed time-resolved photoluminescence (TRPL) measurements to assess the carrier lifetime in the perovskite films with $C_{60}$ (Fig. S24 and Table S2). The MAP/$C_{60}$ sample shows a longer lifetime than the control/$C_{60}$, indicating suppression of the interfacial defects induced by the $C_{60}$ layer. Further analysis of the TRPL data enabled the calculation of the differential lifetimes, indicating rapid electron transfer from the perovskite to the $C_{60}$ for MAP samples (Fig. 3g). Photo-induced charge extraction by linearly increasing voltage (Photo-CELIV) and transient photocurrent (TPC) measurements were also performed to understand the impact of the post-dripping treatment on the charge mobility in the actual cells (Fig. 3h, S25). The photo-CELIV analysis yields a higher charge carrier mobility for the ordered 2D-MAP passivation device $(1.41 \times 10^{-4} \, cm^2 \, V^{-1} \, s^{-1})$ compared to $0.94 \times 10^{-4} \, cm^2 \, V^{-1} \, s^{-1}$ for disordered 2D-MAP-based cells. Lastly, the TPC measurements reveal faster photocurrent decay in cells featuring the ordered 2D-MAP passivation, indicating a more efficient electron transfer/extraction.

The reproducibility of our method was also investigated by evaluating 50 cells from several batches for control and 3D/2D devices with 2D-PEA and 2D-MAP passivation (Figs. 3i, j and S26a, b). In contrast to ordered 2D-MAP passivation-based devices, disordered 2D-PEA-based cells exhibit lower FF and $J_{SC}$ compared to the control devices (Table S3). This notable reduction in FF and $J_{SC}$ is most likely attributed to mismatched energy levels at the 3D/2D/$C_{60}$ interface. Ultraviolet photoemission spectroscopy (UPS) studies (Fig. 3k, Fig. S27, and Table S4) reveal that the conduction band minimum (CBM) of 2D-PEA is higher than that of the control, blocking electron transfer from the perovskite to the $C_{60}$ electron selective contact. Furthermore, DFT calculations show that PEA ligands align perpendicularly to the perovskite surface, potentially hindering electron transport due to the presence of the resistive organic layer[30]. Conversely, MAP ligands are oriented parallel to the 3D perovskite surface and, as such, are not expected to impede electron transport, in line with the above discussion (Fig. S6). It should be noted that even after the post-dripping step with 2D-PEA, an increase in $V_{OC}$ persists compared to control devices, likely due to the PEA-induced passivation on the perovskite surface, which is consistent with the literature (Fig. S28)[23].

We have also tested different organic solvents, such as ethyl acetate and chlorobenzene, as alternative post-dripping solvents (Fig. S29a–d). Although chlorobenzene and ethyl acetate solvents are often used for solvent-dripping treatment of perovskites, images of the ensuing MAP solutions (Fig. S29e) reveal that only IPA can fully dissolve the powder. The low solubility of MAP in these nonpolar solvents is the most likely reason for their inefficacy in facilitating the 2D layer reconstruction on the surface of the perovskite layer. Conversely, as a polar protic solvent, IPA dissolves MAP at a much higher concentration and promotes rapid surface 2D-phase reconstruction observed. Figure S30 shows the impact of ligand concentrations on the cells' operating parameters, from which the optimal formulation was chosen. The optimum concentration for PEAI was approximately 1 mg/mL, consistent with previous findings for inverted PSCs. In the case of MAPCl, using concentrations up to 4 mg/mL induces only a small reduction in the cell's performance. These results indicate that different ligands exhibit varying reactivity with the perovskite layer and require distinct optimization to achieve the formation of a more oriented 2D perovskite phase atop. Furthermore, our findings indicate that the IPA post-dripping strategy demonstrated comparable efficacy for other ligands containing amidino or pyridine groups (Fig. S31). This observation underscores the significance of incorporating additional functional groups into the ligand design and could aid future developments. Finally, our surface passivation strategy was rigorously evaluated for wider bandgap perovskites (1.68 and 1.77 eV). For each material system, the passivation treatment significantly increased the PCEs of the ensuing cells from 19.5% and 16.4% to 21.5% and 18.8%, respectively (Fig. S32, Table S5). These findings highlight the possibility for broader applicability of the proposed surface passivation method for effectively engineering perovskite devices.

## Device stability and ion migration

We evaluated the reliability of our surface passivation method under rigorous stress test conditions by following procedures specified in ISOS protocols[31]. First, we evaluated the thermal stability of control and 3D/2D-passivated cells at a constant temperature of 85 °C in the dark and under an inert atmosphere (ISOS-D-I2, Fig. 4a). The control device exhibited a significant PCE drop of >68% from its initial PCE value after 1200 h. In contrast, the 3D/2D-MAP passivated cells show a significantly improved lifetime, with a PCE drop of ≈14% under identical conditions. On the other hand, the 2D-PEA-passivated devices demonstrated the poorest thermal stability, with an almost 50% PCE drop occurring within the first 200 h. We also evaluated the encapsulated 3D/2D-MAP devices under damp heat conditions at 85 °C and 85% relative humidity (RH). The 3D/2D-MAP devices exhibited almost 82% retention of the initial PCE after 1000 h (ISOS-S-31, Fig. 4b), demonstrating the MAP-treated cells' improved thermal and moisture resilience compared to those treated with the conventional PEA ligand.

Recent studies have shown that at elevated temperatures (≥65 °C), the transformation of the 2D perovskite layer using PEA ligands at the 3D/2D interface leads to rapid cell performance degradation[5,32,33]. To determine the stability of the MAP-induced 3D/2D perovskite layers under similar testing conditions, we employed XRD to analyze the degradation process in samples treated with MAP and PEA. Figure S33b, c presents the structural changes following thermal aging at 85 °C for 48 h in the 3D/2D (MAP) and 3D/2D (PEA) systems, respectively. Notably, the 2D structure characterized by MAP remained discernible, whereas the PEA-based 2D structure was no longer detectable after thermal aging, indicating complete transformation. This observation is consistent with previous reports indicating the possibility of diffusion of PEA cations into bulk perovskites, which might change the structural and optoelectronic properties of the 3D perovskite layer[5]. Using thermal admittance spectroscopy, we further assessed the activation energy ($E_A$) for ion diffusion in control and 3D/2D devices treated with PEA and MAP. Our findings show that MAP treatment increases $E_A$ from 237 to 319 meV (Fig. S34), highlighting its effectiveness in inhibiting ion migration due to stronger binding.

Lastly, we performed outdoor testing of the ensuing PSCs in a hot and humid desert climate in Saudi Arabia. All cells were subjected to continuous operation under daylight illumination and heat cycles from day to night (Fig. 4c). We note that the peak device temperature can reach above 35 °C during the daytime and approximately 20 °C at night, while on a clear sunny day, the intensity of the incident light can easily exceed 1 sun. These conditions represent an ideal test platform to understand the real-world behavior of the developed PSCs. As seen in Fig. 4c, the encapsulated 3D/2D-MAP devices exhibit approx. 75% retention of its initial PCE after 840 h of outdoor testing, demonstrating the potential of these simple post-dripping passivation methods to deliver practical improvements in PSC performance under application-relevant test conditions.

## Discussion

The application of bulky ammonium-based ligands via solution post-treatment of 3D perovskite films is a widely used method for forming 3D/2D perovskite interfaces for highly efficient photovoltaics. Unfortunately, the process leaves residues of unreacted 2D ligands and forms 2D perovskite phases with random orientation and distorted

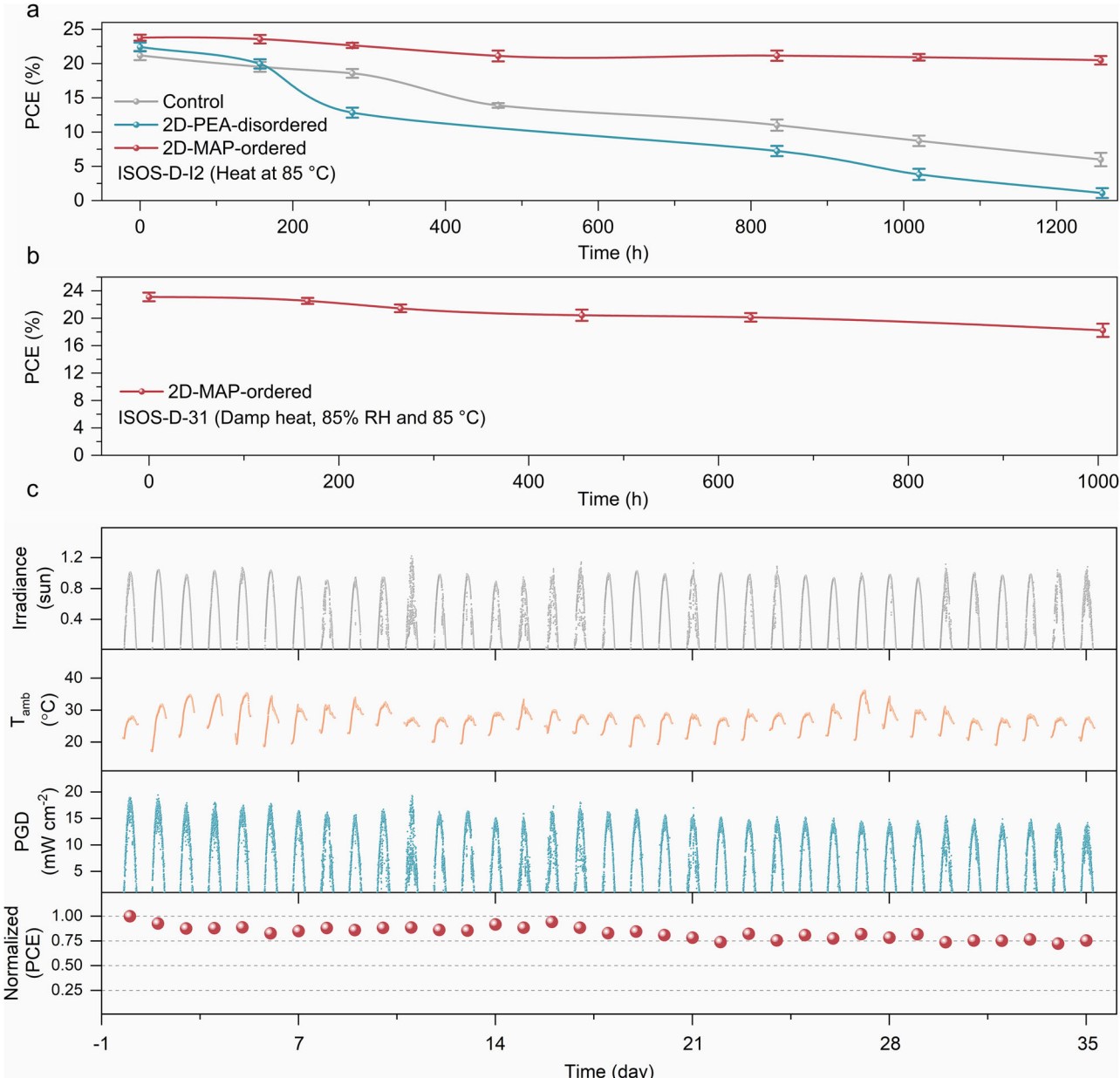

**Fig. 4 | Operating stability of PSCs. a** Thermal aging test for the control, 2D-PEA-disordered-based, and 2D-MAP-ordered-based devices. The cells were maintained at 85 °C in the dark and subjected to intermittent testing inside a nitrogen-filled glove box. Four devices were tested under each condition. **b** Damp heat test (85% R.H., 85 °C) of the encapsulated 2D-MAP-ordered devices. **c** Outdoor stability of a 2D-MAP-ordered device. The outdoor stability measurements were conducted on the KAUST campus in Thuwal, Kingdom of Saudi Arabia, from December 2023 to January 2024 for 35 days.

microstructure, with adverse effects on device performance. We have successfully addressed these issues by introducing a meta-amidinopyridine ligand to generate a phase-pure 2D perovskite ($n = 1$) passivation layer on the surface of a 3D perovskite film. The additional solvent post-dripping step was found to align the initially randomly oriented 2D domains to a highly ordered phase, significantly improving its ferroelectric properties while simultaneously passivating carrier trap states present at the 3D/2D interface. Inverted PSCs featuring this highly oriented and phase-pure 2D perovskite passivation layer exhibited a fill factor of 85.5% and a maximum PCE of over 26% (certified value of 25.44%). Encapsulated cells maintained 82% and 75% of their initial PCE after 1000 h and 840 h, respectively, in damp heat and outdoor tests, demonstrating improved practical durability. The ligand post-dripping step was also successfully applied using dip-coating and blade-coating deposition techniques demonstrating the

scalability of the proposed method. The work highlights the importance of the reactivity of the ligand spacer with the perovskite to create high-quality monolithic 3D/2D perovskite interfaces for developing efficient perovskite photovoltaics.

## Methods

### Materials

[2-(9H-Carbazol-9-yl)ethyl]phosphonic acid (2PACz, >98.0%), lead (II) iodide (PbI$_2$, 99.99%, trace metals basis), lead (II) bromide (PbBr$_2$, 98%, trace metals basis), and *meta*-amidinopyridine hydrochloride (IUPAC: Pyridine-3-carboximidamide Monohydrochloride, MAPCl, 98.0%) were purchased from TCI. N, N-dimethylformamide (DMF, anhydrous, 99.8%), dimethyl sulfoxide (DMSO, anhydrous, 99.9%), and ethyl acetate (anhydrous, 99.8%) were purchased from Sigma-Aldrich. Anhydrous ethanol (EtOH, max. 0.003% H$_2$O, ≥99.8%) was obtained

from VWR Chemicals. Formamidium iodide (FAI, 99.99%), methylammonium iodide (MAI, 99.99%), methylammonium bromide (MABr, 99.99%), and Phenethylammonium iodide (PEAI) were purchased from Greatcell Solar Materials. Cesium iodide (CsI, 99.99%) was purchased from Alfa Aesar. $C_{60}$ was purchased from Nano-C and BCP was purchased from Xi'an Yuri Solar Co., Ltd.

## Perovskite solution preparation

1.53 eV $FA_{0.85}MA_{0.10}Cs_{0.05}Pb(I_{0.98}Br_{0.02})_3$ precursor solution (1.5 M): FAI (0.2193 g), MAI (0.0239 g), CsI (0.0195 g), $PbI_2$ (0.7054 g), and $PbBr_2$ (0.0165 g) were dissolved in 1 mL (DMF: DMSO = 4: 1, volume ratio) and stirred at 25 °C in $N_2$-filled glove box for 12 h. 1.68 eV $FA_{0.70}MA_{0.15}Cs_{0.15}Pb(I_{0.8}Br_{0.2})_3$ precursor solution (1.25 M): FAI (0.1505 g), MABr (0.0210 g), CsI (0.0487 g), $PbI_2$ (0.4466 g), and $PbBr_2$ (0.1032 g) were dissolved in 1 mL (DMF: DMSO = 4: 1, volume ratio) and stirred at 25 °C in $N_2$-filled glove box for 12 h. 1.77 eV $FA_{0.70}MA_{0.15}Cs_{0.15}Pb(I_{0.6}Br_{0.4})_3$ precursor solution (1.25 M): FAI (0.1505 g), MABr (0.0210 g), CsI (0.0487 g), $PbI_2$ (0.2737 g), and $PbBr_2$ (0.2409 g) were dissolved in 1 mL (DMF: DMSO = 4: 1, volume ratio) and stirred at 25 °C in $N_2$-filled glove box for 12 h. The MAPCl and PEAI (2.5 mg mL$^{-1}$ in IPA) solution was stirred at 25 °C in an $N_2$-filled glove box for 12 h.

## Device fabrication

The glass/ITO substrates (Xinyan Technology, 15 Ω cm$^{-1}$, and 2.54 cm × 2.54 cm) were cleaned using dilute Extran 300 detergent solution, deionized water, acetone, and isopropanol and then dried in an $N_2$ flow before being treated with $O_3$ plasma for 15 min. Then, the self-assembled monolayer 2PACz or mixture of 2PACz and 4-hydroxybenzylamine solution (1 M in ethanol) was spin-coated on glass/ITO substrates at 5000 r.p.m. for 30 s followed by annealing at 100 °C for 10 min[14]. For 1.53 eV perovskite film fabrication, a spin-coating procedure with 1000 r.p.m. for 10 s and 4000 r.p.m. for 40 s was adopted to prepare perovskite films. Ethyl acetate or anisole (200 μL) was dropped on the spinning substrate at the last 5 s of the second spin-coating step followed by annealing at 100 °C for 30 min in $N_2$ conditions. For 1.68 eV perovskite film fabrication, a spin-coating procedure with 2000 r.p.m. for 10 s and 5000 r.p.m. for 30 s was adopted to prepare perovskite films. Ethyl acetate (300 μL) was dropped on the spinning substrate at the last 10 s of the second spin-coating step followed by annealing at 100 °C for 30 min in $N_2$ conditions. For 1.77 eV perovskite film fabrication, a spin-coating procedure with 2000 r.p.m. for 10 s and 5000 r.p.m. for 40 s was adopted to prepare perovskite films. Chlorobenzene (200 μL) was dropped on the spinning substrate at the last 20 s of the second spin-coating step followed by annealing at 100 °C for 30 min in $N_2$ conditions.

A spin-coating procedure with 5000 r.p.m. for 30 s was adopted to prepare 2D layers. 70 μL MAPCl or PEAI solution was dropped on the spinning perovskite films followed by annealing at 80 °C for 5 min in $N_2$ conditions. The treated films were then thoroughly washed with IPA several times (i.e., 1–10 times) after cooling to get 2D-MAP-ordered perovskite films. We also employed scalable fabrication techniques to validate the effect of IPA post-dripping. For the blade-coating method, IPA post-dripping was performed by blade-coating on substrates at 25 °C at a speed of 1.5 m/min in a glove box. A nitrogen ($N_2$) flow was applied after the blade to accelerate the evaporation of IPA. For the dip-coating method, IPA post-dripping was carried out by immersing the 3D/2D-MAP-disordered film in a beaker of IPA for approximately 3 s, followed by drying the film with $N_2$ flow to remove residual IPA in the glove box. The devices were completed by thermal deposition of $C_{60}$ (30 nm), BCP (7 nm), and silver (100 nm) at a vacuum of <4 × 10$^{-6}$ torr. The device area by evaporation was 0.1 cm$^2$. Unless otherwise stated, the devices were masked with metal aperture masks (0.063 cm$^2$) during the $J$−$V$ measurement.

## Characterizations

**GIWAXS**. Two-dimensional grazing-incidence wide-angle X-ray scattering (GIWAXS) measurements were conducted at the 9A U-SAXS beamline of the Pohang Accelerator Laboratory (PAL) in the Republic of Korea. These measurements were performed with an X-ray beam energy of 11.07 KeV and an incident angle of 0.10° and 0.50°.

**SEM**. The surface morphologies of perovskite films were characterized by SEM (Zeiss, Auriga).

**AFM and KPFM**. We utilized a Digital Instruments MultimodeAFM (Veeco Metrology Group) and a Bruker Head to conduct topographic and KPFM potential mappings. The SCM-PIT V2 AFM tip was employed, which was manufactured using an RFESP-75 AFM probe and coated with Reflective PtIr (Platinum-Iridium). The AFM tip was composed of Antimony-doped Silicon and featured a resistivity ranging from 0.01 to 0.025 Ω cm. The tip was rectangular in shape and boasted nominal frequency and stiffness values of 75 KHz and 4 N/m, respectively. After calibrating the WF of the probe tip using a gold reference substrate, we calculated the WF of the probe tip, which yielded a value of approximately 4900 mV.

**PFM**. Asylum research atomic force microscope was used for piezoelectric performance of perovskite films.

**HR-STEM and FIB**. For the high-resolution scanning transmission electron microscopy (HR-STEM)-based study, a cross-sectional electron-transparent lamella was prepared using a focused ion beam (FIB) equipped scanning electron microscope (SEM-FIB Helios G5 DualBeam, FEI) with the assistance of an EasyLift nanomanipulator and a Gallium (Ga) ion source. To protect the region of interest during FIB processing, two types of protective coatings were deposited: first, a 0.5 μm layer of W coating was applied using the e-beam, followed by a 3 μm layer of W coating deposited by the ion beam for final protection. The ion beam milling procedure was carried out step by step, with a beam current of 2.4 nA, 0.44 nA, 0.26 nA, 0.045 nA, and 0.025 nA, gradually reducing it from 30 kV to 5 kV, in order to cut and thin down the lamella to 60 nm while minimizing ion beam damage. A low current cleaning process (5–2 kV, 81–28 pA) was performed to eliminate potential contamination. Scanning Transmission Electron Microscopy (STEM)-based experiments were conducted using a Cs Probe-corrected Thermo Fisher Titan 60–300 Cubed TEM microscope operating at 80 kV. TEM data processing was carried out using Gatan™ Digital Micrograph and the Thermo Scientific™ Velox suites.

**QFLS**. Absolute PL imaging measurements were conducted using hyperspectral fluorescence microscopy (Photon etc. IMA) equipped with a 20x magnification microscope. The samples were encapsulated using ultra-thin glass and exposed to 532 nm lasers, which were calibrated to 1 sun condition by adjusting the power and optical density. The data-collecting procedure followed ref. 34. Thus, the QFLS was converted from absolute PL by using home-built MATLAB code[35].

**UPS and IPES**. The electron spectroscopy measurements of ultraviolet photoelectron spectroscopy (UPS) and low energy inverse photoemission spectroscopy (LE-IPES) were conducted in a single UHV ScientaOmicron system at 10$^{-9}$ mbar. The surface work function and valence band maximum were studied by UPS with a vacuum ultraviolet unfiltered He (1) (21.22 eV) source (focus). The samples were biased by 10 eV to observe the secondary electron cutoff. The photoelectrons were collected at an angle of 80° between the sample and analyzer, with a normal electron take-off angle. The constant analyzer pass energy (CAE) was 3 eV for the valence band region and for the secondary electron cutoff. IPES was conducted in a home-built chamber

consisting of an electron source (Staib) operating at 20–30 eV with an energy dispersion of 0.25 eV directed normal to the sample. The sample was biased by 20 eV. All extraneous light was eliminated during measurements by covering port windows and sealing the detector. Plots of UPS and IPES are constructed considering the shared EF position and arbitrarily adjusted in intensity.

**TPC and Photo-CELIV measurements.** TPC and photo-induced charge extraction linear increasing voltage (Photo-CELIV) measurements of full device stacks were performed by the Fluxim PAIOS system. PAIOS exploits a 19-first function generator to control the light source (a white LED with a rise/fall time of 100 ns), and a second function generator to control the applied voltage bias. The output current is measured through a trans-impedance amplifier. The white LED (100 mA) has been calibrated with a certified silicon reference (RERA Solutions).

**NMR.** $^1$H NMR spectra were acquired using an AVANCE III 500 nuclear magnetic resonance spectrometer from Bruker, Germany, which operates at a frequency of 500 MHz and a temperature of 25 °C.

**DFT.** The results were obtained with the Density Functional Theory (DFT) code Vienna Ab Initio Simulation Package (VASP)[36] with a plane-wave basis (and an energy cutoff of 500 eV), projector augmented waves (PAW)[37] and the generalized gradient approximation (GGA) Perdew-Burke-Ernzerhof[38] exchange-correlation (xc) functional. The non-bonding van der Waals interactions were taken into account within the so-called DFT-D3 scheme[39]. Structures were rendered with the software VESTA[40]. The substrate was chosen to be a slab with 4 layers of the alpha phase of formamidinum (FA) lead iodine (FAPbI$_3$) with (110) nonpolar terminations. We have also modeled the interactions of AP molecules with FA$_{1-x-y}$MA$_x$Cs$_y$PbI$_3$, where $x = y = 0.0625$ and MA stands for methylammonium. The lateral dimensions of the supercell of the slab were 18.56 Å × 12.77 Å. Using the Gaussian 09 software, we computed the electrostatic potentials and dipole moments of the ligands. The calculations were performed at the B3LYP level with a def2-TZVP basis set and included DFT-D3 dispersion corrections.

**Device characterizations.** The $J-V$ performance of the perovskite solar cells was analyzed using a Keithley 2400 source meter under N$_2$ conditions at room temperature, and the illumination intensity was 100 mW cm$^{-2}$ (AM 1.5G Abet Technologies Sun 3000 solar simulator). The scan range was 1.2 V → −0.1 V, and the bias step was 0.01 V. The power output of the lamp was calibrated using a standard silicon cell. The EQE was characterized on a commercial Newport EQE system under an N$_2$ environment. The chopped monochromatic light beam was focused entirely on the active area of the solar cells.

**Encapsulation and stability tests**
The device is covered by two glass layers and two encapsulant sheets. We utilize edge sealant (PVS 101) to hinder moisture entry from the sides of the device. The stack is vacuum laminated using an industrial laminator (Ecolam5 Ecoprogetti) at 120 °C for 20 min. All procedures were conducted in the atmosphere. The perovskite devices remain s during the encapsulation process.

**Reporting summary**
Further information on research design is available in the Nature Portfolio Reporting Summary linked to this article.

## Data availability
All relevant data are presented in the in the Supplementary Information/Source Data file. Source data are provided with this paper.

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

## Acknowledgements

This publication is based upon work supported by the King Abdullah University of Science and Technology (KAUST) Office of Sponsored Research (OSR) under Award Nos. OSR-2018-CARF/CCF-3079 and OSR-2019-CRG8-4095. The DFT calculations used computational time granted from the GRNET S.A. in the HPC facility ARIS under projects 13006 and 15002. C.X. is thankful for the support from the National Natural Science Foundation of China (52302327). R.A. acknowledges financial support from the University Development Fund (UDF-01003793), the Outstanding Young Talents Program (Overseas) of the National Natural Science Foundation of China (NSFC, 2023), and Shenzhen Peacock Program (C Talent). H.Y.W. acknowledges the financial support from national research foundation of Korea (2019R1A6A1A11044070 and RS-2024-00334832). We also extend our thanks to Chuang Ma, Tianqi Niu, Shanshan Zhang, Asmat Ullah, and Fuzong Xu for their valuable discussions of this research, to Anil Reddy Pininti for preparing the aperture mask, and to Yiwei Hu from Shiyanjia Lab (www.shiyanjia.com) for assisting with the XPS analysis.

## Author contributions

X.C. and R.A. conceived the project idea, wrote the initial manuscript, and fabricated and characterized the perovskite solar cells. T.Y., N.W., and K.Z. helped with solar cell certification. H.X. and C.X. measured and analyzed the cross-sectional KPFM samples. F.H.I. and H.F. assisted in measuring the activation energy. P.D. contributed to the UPS and IPES measurements. M.J.H. and Z.L. conducted the TPC and Photo-CELIV measurements. S.Y.J. and H.Y.W. performed GIWAXS. D.S.U. and R.A. were involved in PL and QFLS measurement and analysis. B.V. conducted FIB and TEM measurements and analyses. L.T. performed the DFT calculations and analysis. Y.Y.Y. helped with the ligands design talk. M.M. and R.A. carried out encapsulation and stability tests. A.P. conducted the AFM and KPFM measurements. R.A., S.D.W., and M.H. supervised the project and secured the funding. T.D.A. contributed to the ideas, material analysis, supervised the project, and secured funding. All authors contributed to the writing of the manuscript.

## Competing interests

The authors declare no competing interests.
