## [Transparent Peer Review file · Nature Communications]

Solvent-dripping modulated 3D/2D heterostructures for high-performance perovskite solar cells

Corresponding Author: Professor Thomas Anthopoulos

Version 0:

Reviewer comments:

Reviewer #1

(Remarks to the Author)

The manuscript titled "Solvent-dripping modulated 3D/2D heterostructures for high-performance perovskite solar cells" has undergone a thorough review. In this manuscript, the authors introduce a meta-amidinopyridine ligand via a solvent post-dripping step to produce a highly ordered 2D perovskite phase atop a pre-deposited 3D perovskite film. The manuscript is well-prepared. Consequently, I recommend its publication in Nature Communications, contingent upon the implementation of the improvements listed below.

1. The PbI₂-rich domains in the removed 2D-PEA sample are larger than those in the control sample, regardless of whether post-dripping was applied. What accounts for this difference?
2. How did the post-dripping process reorient the 2D-MAP perovskite to achieve $n=1$ as shown in Figure 1c? The reason of pure 2D-MAP perovskite with $n=1$ should be elucidated.
3. In the DFT section, why did the authors choose different halide ions?
4. Authors should provide the process of optimizing the 2D perovskite concentration.
5. Why is the QFLS value of the 2D-PEA-disordered with C60 sample lower than that of the control sample, whereas the Voc is higher?
6. The authors need to add the recent published works, such as, silver coordination-induced n-doping of PCBM for stable and efficient inverted perovskite solar cells; Phosphonate Diacid Molecule Induced Crystallization Manipulation and Defect Passivation for High-Performance Inverted MA-Free Perovskite Solar Cells.
7. The characterization techniques of TPC and Photo-CELIV should be described in detail.
8. The industry requires larger perovskite PV modules. It is recommended that the authors demonstrate the compatibility of their method with scalable fabrication techniques, including blade coating, slot-die coating, and spray coating.

Reviewer #2

(Remarks to the Author)

The paper entitled "Solvent-dripping modulated 3D/2D heterostructures for high-performance perovskite solar cells" investigates the concept of utilizing a post-dripping step to produce a highly ordered 2D perovskite phase on the surface. It demonstrates that reconstructed 2D/3D perovskite heterointerface showed reduced energetic disorder. However, to fully realize the potential of this concept and its applicability, a more comprehensive understanding of the underlying mechanisms is crucial. The manuscript falls short in terms of deep discussion and interpretation of the data, so substantial revisions should be done before its consideration for publication in Nature Communications.

1. Is there a typo in the last picture of Figure 1a? 2D-MAP-ordered or 2D-PEA-ordered.
2. The authors consider the order $n=1$ 2D perovskite is favorable. However, previous works (Science 2022, 376, 73–77; Nat. Photon. 2022, 16, 352-358) expressed opposite views for the phase structures. Can the author comment on it?
3. The authors claim that post-dripping treatment (3D/2D-MAP-ordered films) leads to the reorganization of the 2D perovskite phase on the surface and potentially within the 3D perovskite layer. What is the reason for this and what has happened in this treatment. This is the key point of this work. Detailed analysis with comprehensive discussion is required.
4. Following the question, why do the different ligands (PEA and MAP) lead to contrasting results? Moreover, why do the authors choose these two ligands? The chemical structures are very different. PEA like MA, MAP like FA. So, the story is getting too complicated.

5. What is the thickness of the 2D capping layer? From the TEM results, we estimated the thickness of the 2D phase is around 40 nm. According to the reference (Science 2022, 377, 1425–1430), the thickness should be optimized in the p-i-n configuration. Moreover, the orientation of the 2D perovskite is not favorable for charge transfer.
6. I cannot agree with the hypothesis of Figure 3g. Transient absorption (TA) measurements would reflect the sign of the whole film rather than the interface charge transfer. In contrast, I suggest the author measure the TA for the 2D/3D perovskite with different treatments (materials and methods). Thus, the authors should analyze the charge transfer between different phases based on the PB characteristics.
7. Energy alignment schemes should be given in the main part. In the SI, I found that the over-reduction of the WFs would be harmful.
8. Since QFLS has been given, I encourage the author to present the VOC-loss analysis for the direct comparison between different treatments.
9. Could the post-dripping be used in the large-area solution-processing? It is very important for the application of this method. Otherwise, the results of 1.0 cm² devices are not attractive.

Reviewer #3

(Remarks to the Author)

Authors report the use of a meta-amidinopyridine (MAP) ligand through a solvent post-dripping step to produce a highly ordered 2D perovskite phase on the surface of 3D perovskite film. The resulting reconstructed 2D/3D perovskite heterointerface showed reduced energetic disorder and enhanced performance compared to the control and randomly oriented 2D/3D perovskite sample. PSCs made with this ordered 2D-phase perovskite achieved a maximum power conversion efficiency (PCE) of 26.05% (a certified value of 25.44%). During damp heat and outdoor tests were also performed to demonstrate the improve stability. The paper might be accepted after addressing the following issues.

1. The qz peak of 2D for 2D-MAP in GIWAX shifts significantly after IPA dripping. Thus, the statement “are expected to react less with the post-dripping solvent or have stronger binding to the inorganic octahedral perovskite matrices” is not entirely accurate—it’s not about reacting less but rather about reconstruction. Also, in Fig. 1a, the x-axis should be labeled as qxy instead of qz. Additionally, is there any difference observed in the XRD?
2. TEM images in Fig 1b are not very convincing because perovskite materials easily decompose under FIB milling process (unless using cryo-FIB) and the highlighted area might be PbI₂ instead of 2D perovskite.
3. The N 1s spectra in MAP need fitting, as there are two different N bonds (N in FA, MA, and pyridinic nitrogen atoms). Therefore, the statement that N shifts significantly is not strongly supported.
4. Fig. 3g, the authors mention that these are TA results; however, they do not specify the exact wavelength for these dynamics. Additionally, the time scale in Fig. 3g is in ns, while in Table S2, it is in ps. How can this data be fitted in such a manner? Could it be TRPL?
5. The authors mention “reveal that the conduction band minimum (CBM) of 2D-PEA is higher than that of the control, blocking electron transfer from the perovskite to the C60 electron selective contact”, while I can’t find the CBM data for 2D-PEA, only for control and 2D-MAP in Fig. S17. Also, based on the UPS results, the calculated energy level for control and 2D-MAP is -3.94/-5.5 eV and -3.84/ -5.51 eV. Based on this, the 2D-MAP seems to block the electron extraction of control. Can the authors elaborate more?
6. Regarding the use of novel ligand and 2D structures in 2D/3D perovskite solar cells, some recent work on conjugated molecular design (e.g., Science Advances 2023, 9, eadg003) could be cited.

Version 1:

Reviewer comments:

Reviewer #1

(Remarks to the Author)

All the comments have been addressed well, accept.

Reviewer #2

(Remarks to the Author)

The authors have responded to the concerns in the revised manuscript. However, some of the critical questions have not been settled.

For Comment 3, the authors suggested that post-dripping process helps to remove excess ligands and reorient the 2D perovskite domains from a random orientation to a highly ordered layer. I cannot follow this explanation. IPA treatment would cause some dissolution of the perovskite surface. Besides, why the ordered structure is thermodynamically more stable? And why the 2D film is uneven?

For Comment 4, my question is that PEA and MAP are obviously not comparable in the case of structural characteristics. I suggest re-selecting the reference object.

For Comment 6, if the 2D film is highly ordered, the 2D characteristics should be identified. Moreover, I found the characteristic PB peak of 3D perovskite widens when using 2D-ordered. As we know, a wide FWHM indicates an increased defect density, which doesn't make any sense. Pleased explain it.

Reviewer #3

(Remarks to the Author)

My questions have been well-addressed and in my opinion it can be accepted now.

Version 2:

Reviewer comments:

Reviewer #2

(Remarks to the Author)

I have no further question about this work.

REPLY to REVIEWERS' COMMENTS

We are pleased to submit the revised version of our manuscript (NCOMMS-24-48452-T) for reconsideration in *Nature Communications*. We are very grateful to the Reviewers for reviewing our work and providing constructive feedback. Next, we address all points raised in turn. All revisions are *highlighted* in the revised manuscript files for convenience.

Kind regards,

Prof. Thomas Anthopoulos & co-authors

Reviewer #1 (Remarks to the Author):

Comment: The manuscript titled "Solvent-dripping modulated 3D/2D heterostructures for high-performance perovskite solar cells" has undergone a thorough review. In this manuscript, the authors introduce a meta-amidinopyridine ligand via a solvent post-dripping step to produce a highly ordered 2D perovskite phase atop a pre-deposited 3D perovskite film. The manuscript is well-prepared. Consequently, I recommend its publication in *Nature Communications*, contingent upon the implementation of the improvements listed below.

Response: We thank the Reviewer for the positive feedback and appreciate the recommendation for publication in *Nature Communications*. Next we address all points raised in turn.

Comment: 1. The PbI_2 -rich domains in the removed 2D-PEA sample are larger than those in the control sample, regardless of whether post-dripping was applied. What accounts for this difference?

Response: Indeed, the PbI_2 -rich region on the perovskite film surface in the removed 2D-PEA sample appears more extensive than in the control sample, as evidenced by **Fig. S3d** and **3f** in the top-view SEM images. This is because most PEA ligands in 2D-PEA dissolve and are removed during post-dripping, resulting in large areas of residual PbI_2 domain. The primary reasons for this phenomenon are the weak interaction between the PEA ligands and the inorganic octahedral sheet, as well as the higher solubility of the PEA ligands in IPA. In contrast, our 2D-MAP ligand showed minimal residual PbI_2 after post-dripping, indicating that MAP ligands formed stronger bonds with the inorganic octahedral sheet, facilitating the reorientation of the 2D-MAP layer into an ordered structure. We also compared the solubility of PEAI and MAPCl in IPA. At a concentration of 10 mg/mL, MAPCl reached its saturation point, whereas PEAI remained unsaturated, with the solution remaining clear even at a higher concentration (40 mg/mL). This confirms that ligand solubility in the post-dripping solvent plays a key role in the removal of the 2D ligand and the formation of residual PbI_2 .

Fig. S3g: Solubility test of PEAI and MAPCl in IPA.

In response, we have revised the text on page 5:

“However, after the post-dripping step, the morphology returns to that of the control perovskite sample accompanied by the presence of large PbI_2 -rich domains (bright white). The higher

solubility of the PEA ligand in IPA compared to that of the MAP ligand facilitated surface reconstruction (see Fig. S3g), involving the removal of 2D layer and formation of residual PbI_2 .” Additionally, we added Fig. S3g in SI file.

Comment: 2. How did the post-dripping process reorient the 2D-MAP perovskite to achieve $n = 1$ as shown in Figure 1c? The reason of pure 2D-MAP perovskite with $n = 1$ should be elucidated. **Response:** This is a critical question and we appreciate the opportunity to clarify the reorientation process of the 2D perovskite layer and phase purity.

Generally, the solution post-treatment of 2D ligands leads to the formation of excess ligands and an uncontrollable mixed phase of 2D perovskites, including an intermediate phase, limiting the 2D arrangement and orientation. Our post-dripping process helps to remove excess ligands and reorient the 2D perovskite domains from a random orientation to a highly ordered layer. The process involves post-dripping the film with IPA multiple times. This post-dripping helps remove excess MAP ligands and allows for the reorientation of the 2D perovskite domains. Initially, the 2D perovskite has a random orientation, but through this process, it becomes more ordered in the z-direction (parallel orientation), which is thermodynamically more stable.¹⁻³ GIWAXS (Grazing-Incidence Wide-Angle X-ray Scattering) measurements were used to confirm this mechanism, as shown in the updated GIWAXS data in Fig. 1a. The results showed that after the first IPA post-dripping, the 2D signal weakened but remained disordered. However, after the 3rd and 4th post-dripping, the 2D layer became more ordered in the z-direction. Importantly, 2D-MAP maintained its phase purity throughout this process. Further post-dripping (up to 10 times) led to the weakening of the 2D-MAP-ordered peak and the strengthening of the residual PbI_2 peak. This indicates that the MAP ligand remains strongly bound to the inorganic PbI_6 octahedral framework even after multiple post-dripping. In contrast, 2D-PEA was completely removed after the first IPA post-dripping.

To clarify the above mentioned process, we have added a schematic, shown as revised Fig. 1c, depicting the reorientation process of the 2D-MAP involving ligand dissolution and removal followed by reorientation of the 2D-MAP.

In response we added the revised Fig. 1a and 1d.

Fig. 1a: 2D GIWAXS images of perovskite films with 0.1° X-ray incidence angles.

Fig. 1d: Schematic illustrating the reorientation of the 2D-MAP perovskite during and after the post-dripping process.

On the other hand, the formation of phase-pure or quasi-2D perovskites depends on the ligand design and fabrication process. For ligand design, most conventional 2D ligands such as phenethylammonium (PEA) or butylammonium (BA) will form quasi 2D perovskite with multiple phases $n = 1, 2, 3, \text{etc.}$ In contrast, the 2D ligand with an additional functional group forms additional hydrogen bonding that induces phase-pure 2D perovskite.^{4,5} Our meta-amidinopyridine (MAP) ligand has pyridine and amidino groups that can interact with Pb and halide sites to minimize the penetration of the ligand and induce phase-pure 2D perovskite with $n = 1$. The above part/response has also been integrated into the revised manuscript.

Comment: 3. In the DFT section, why did the authors choose different halide ions?

Response: We thank the Reviewer for this question. In this work, we did not focus on different halide ions but on the cation molecule to investigate the orientation of the cation on the inorganic octahedral sheet using DFT. We compared the PEAI ligand as the commonly used ligand with the MAPCl ligand to form 2D perovskite passivation. We also tested the MAP ligand with iodide ions, but its performance was lower than that of the MAPCl ligand, which might be due to the better passivation effect of chlorine than iodide. Therefore, to realistically understand the experimental results with DFT, we compared PEAI with MAPCl in our calculations.

Comment: 4. Authors should provide the process of optimizing the 2D perovskite concentration.

Response: We appreciate the Reviewer's valuable comment. To determine the optimal concentration of 2D perovskite, we initially conducted a series of experiments. These tests involved altering the concentration of the 2D ligand solution while maintaining consistency in the other parameters. Our experiments explored concentrations ranging from 1.0 mg/mL to 4.0 mg/mL, and we observed the effects on device performance. Our findings revealed that devices achieved the highest PCE when the MAPCl concentration was 2.5 mg/mL following post-dripping. In contrast, for the PEA ligand, a lower concentration of approximately 1 mg/mL without post-dripping proved optimal, as the post-dripping process tends to remove the 2D layer or PEA ligand. These results indicate that different ligands exhibit varying reactivity and require distinct optimal concentrations to effectively interact with the 3D perovskite layer and form 2D perovskite. We have also added the device statistics of different concentrations of PEA and MAP ligands for better clarity. In response, we have added **Fig. S28**.

Fig. S28: PCE, V_{OC} , fill factor, and J_{SC} statistics of PSCs with different ligand concentrations of (a-d) 2D-MAP-ordered and (e-h) 2D-PEA-disordered. The statistics are obtained from 12 cells of each condition from different batches.

Comment: 5. Why is the QFLS value of the 2D-PEA-disordered with C₆₀ sample lower than that of the control sample, whereas the V_{OC} is higher?

Response: The QFLS mapping of 2D-PEA-disordered was found to be more inhomogeneous than that of the other samples, especially after C₆₀ deposition; therefore, we might put the previous QFLS value inaccurately for 2D-PEA with C₆₀. The inhomogeneity of the QFLS mapping on 2D-PEA-disordered can originate from the non-uniform 2D layer distribution and passivation of 2D-PEA quality. For this reason, we repeated our measurement for different samples and spots to obtain the accurate average QFLS value of each sample and concluded that the average QFLS value of 2D-PEA-ordered is slightly higher than that of the control, consistent with the V_{OC} value of the devices.

In response, we revised **Fig. 3e** and **f**.

Fig. 3e,f: QFLS distribution of the perovskite films without (e) and with (f) C₆₀.

Comment: 6. The authors need to add the recent published works, such as, silver coordination-induced n-doping of PCBM for stable and efficient inverted perovskite solar cells; Phosphonate Diacid Molecule Induced Crystallization Manipulation and Defect Passivation for High-Performance Inverted MA-Free Perovskite Solar Cells.

Response: Thank you for your suggestion. We added both of these studies into the revised manuscript. Please see references 9 and 10.

Comment: 7. The characterization techniques of TPC and Photo-CELIV should be described in detail.

Response: We have provided a detailed description of the TPC and Photo-CELIV characterization techniques in the experimental section of the revised manuscript.

Added detail experiment in method section.

“TPC and Photo-CELIV measurements: TPC and photo-induced charge extraction linear increasing voltage (Photo-CELIV) measurements of full device stacks were performed by the Fluxim PAIOS system. PAIOS exploits a 19 first function generator to control the light source (a white LED with a rise/fall time of 100 ns), and a second function generator to control the applied voltage bias. The output current is measured through a trans-impedance amplifier. The white LED (100 mA) has been calibrated with a certified silicon reference (RERA Solutions).^{6”}

Comment: 8. The industry requires larger perovskite PV modules. It is recommended that the authors demonstrate the compatibility of their method with scalable fabrication techniques, including blade coating, slot-die coating, and spray coating.

Response: We agree that demonstrating compatibility with scalable manufacturing techniques is essential for industrial applications. Our primary objective is to modify the 2D perovskite

passivation domain by implementing a straightforward post-dripping step. This process eliminated the excess unbound ligands in the film and dissolved the 2D ligand, thereby reorienting the 2D passivation layer to become more ordered in the z-direction (parallel orientation). Our post-dripping method also offers a scalability aspect. To demonstrate this, we tested IPA post-dripping on 2D-MAP passivated films using the blade coating or dipping method, which is scalable for large areas. The results showed that post-dripping using blade coating or the dipping method improved the 1-cm² device performance compared to cells without post-dripping, indicating the method's promise for large-scale fabrication compatibility. In response, we added blade- and dip-coating of IPA on 1-cm² devices in **Fig. S19** and **S20**.

Fig. S19: Schematic diagram of blade-coating and dip-coating methods. We employed scalable fabrication techniques to validate the effect of IPA post-dripping. For the blade-coating method, IPA post-dripping was performed by blade-coating on substrates at 25°C at a speed of 1.5 m/min in a glove box. A nitrogen (N₂) flow was applied after the blade to accelerate the evaporation of IPA. For the dip-coating method, IPA post-dripping was carried out by immersing the 3D/2D-MAP-disordered film in a beaker of IPA for approximately 3 seconds, followed by drying the film with N₂ flow to remove residual IPA in the glove box.

Fig. S20: Statistical comparison of photovoltaic parameters (V_{oc} , J_{sc} , FF, and PCE) of 2D-MAP-disordered and 2D-MAP-ordered-based devices by blade-coating and dip-coating of IPA on 3D/2D-MAP films (1.0 cm²).

=====

Reviewer #2 (Remarks to the Author):

The paper entitled "Solvent-dripping modulated 3D/2D heterostructures for high-performance perovskite solar cells" investigates the concept of utilizing a post-dripping step to produce a highly ordered 2D perovskite phase on the surface. It demonstrates that reconstructed 2D/3D perovskite heterointerface showed reduced energetic disorder. However, to fully realize the potential of this concept and its applicability, a more comprehensive understanding of the underlying mechanisms is crucial. The manuscript falls short in terms of deep discussion and interpretation of the data, so substantial revisions should be done before its consideration for publication in Nature Communications.

Response: We would like to thank to the Reviewer for the constructive comments and suggestions to further improve the quality of our manuscript.

Comment: 1. Is there a typo in the last picture of **Figure 1a**? 2D-MAP-ordered or 2D-PEA-ordered.

Response: We appreciate for highlighting this error. We mistakenly labelled 2D-MAP-ordered as 2D-PEA-ordered and corrected this error in Fig. 1a in the revised manuscript file.

Comment: Comment: 2. The authors consider the order $n = 1$ 2D perovskite is favorable. However, previous works (Science 2022, 376, 73–77; Nat. Photon. 2022, 16, 352-358) expressed opposite views for the phase structures. Can the author comment on it?

Response: We appreciate the Reviewer's comment. It is true that having the narrow bandgap of 2D perovskites is desirable to minimize the electron-blocking alignment on 3D/2D heterojunctions in p-i-n configuration. However, this also depends on the properties and crystal structure of the 2D perovskites atop the 3D perovskite film. For example, the coverage of 2D perovskite affects charge transport. In general, the conformal planar 2D layer likely blocks electron transfer from the 3D perovskite to the C₆₀ ETL. In contrast, our 2D-MAP is expected to be discontinuous, which minimizes electron blocking issues, as shown in the cross-sectional TEM images (**Fig. 1c** and **Fig. S4c**). Additionally, the VBM of 2D-MAP with pure $n = 1$ exhibit a deeper level (-3.84 eV) while for 2D-PEA has a higher position at -3.23 eV, minimizing the energetic offset (0.09 eV for 2D-MAP and 0.70 eV for common 2D-PEA with the same $n = 1$ phase). Furthermore, the 2D structure formed by MAP exhibits ferroelectric properties that are absent in conventional 2D perovskites. The built-in electric field from the ferroelectric phase enhances charge transport, offering a distinct advantage over non-ferroelectric 2D perovskites.⁷ These are key differences that we believe are at play and determine the overall device performance. We have tried to clarify these points in the revised manuscript.

Comment: 3. The authors claim that post-dripping treatment (3D/2D-MAP-ordered films) leads to the reorganization of the 2D perovskite phase on the surface and potentially within the 3D perovskite layer. What is the reason for this and what has happened in this treatment. This is the key point of this work. Detailed analysis with comprehensive discussion is required.

Response: This is a critical question which was also raised by Reviewer #1 above. Please see our response to point number 2 for Reviewer #1 above. In brief, post-dripping helps reorient the 2D perovskite domains from a random orientation to a highly ordered layer. Conventionally, solution post-treatment of ammonium-based ligand will form an excess organic ligand and impurity, limiting the 2D orientation arrangement and quality. Our post-dripping process helps to remove excess ligands and reorient the 2D perovskite domains from a random orientation to a highly ordered layer. The process involves post-dripping the film with IPA multiple times. This post-dripping helps remove excess MAP ligands and allows for the reorientation of the 2D perovskite

domains. Initially, the 2D perovskite has a random orientation, but through this process, it becomes more ordered in the z-direction (parallel orientation), which is thermodynamically more stable,¹⁻³ as confirmed by newly added GIWAXS data in **Fig. 1a**. This suggests that after the 3rd and 4th IPA post-dripping, the 2D layer became more ordered in the z-direction. We also added a schematic to understand the reorientation process of the 2D-MAP involving ligand dissolution and removal followed by reorientation of the 2D-MAP, as illustrated in the revised **Fig. 1d**.

Fig. 1a and **1d** were added to the revised manuscript along with the relevant text.

Fig. 1a: 2D GIWAXS images of perovskite films with 0.1° X-ray incidence angles.

Fig. 1d: Schematic illustrating the reorientation of the 2D-MAP perovskite during and after the post-dripping process. The above part has also been integrated in the revised manuscript.

Comment: 4. Following the question, why do the different ligands (PEA and MAP) lead to contrasting results? Moreover, why do the authors choose these two ligands? The chemical structures are very different. PEA like MA, MAP like FA. So, the story is getting too complicated.

Response: We thank the Reviewer for the question. PEAI is a commonly used ligand for surface passivation in perovskite solar cells and serves as a foundation for the development of numerous other 2D ligands.⁸⁻¹¹ Thus, our choice of PEA as a representative ligand for this study was justified by its extensive use in the field. However, recent research has highlighted the stability limitations associated with PEA passivation.¹²⁻¹⁴ To enhance both the PCE and stability, it is crucial to identify and design an appropriate ligand for creating 2D perovskite passivation with a rigid and well-ordered crystal structure. In our study, we introduced the MAP ligand, which has rarely been explored in 3D/2D heterojunctions on the electron-selective contact side. Unlike PEA, which possesses only a single ammonium group, MAP features additional functional groups, such as amidino and pyridine, capable of interacting with the octahedral perovskite networks. Thus, enhancement of the chemical bonding and passivation effects further improve the performance and stability of PSCs. These are the primary reasons for choosing the specific ligands.

Comment: 5. What is the thickness of the 2D capping layer? From the TEM results, we estimated the thickness of the 2D phase is around 40 nm. According to the reference (Science 2022, 377, 1425–1430), the thickness should be optimized in the p-i-n configuration. Moreover, the orientation of the 2D perovskite is not favorable for charge transfer.

Response: We thank the Reviewer for the question. It is correct that the 2D perovskite has the optimum thickness and often a thinner layer for p-i-n cells is preferred. In addition, the 2D perovskite orientation and distribution also play an important role in passivation and charge transfer. To our knowledge, the roles of the latter parameters have not been studied adequately. From the TEM images presented, and despite our extensive efforts, we failed to verify the presence of a continuous 2D layer on top of 3D perovskite. Instead, we see that the 2D-MAP distribution is uneven/discontinuous. We believe the latter characteristic helps to minimize charge blocking, allowing for optimal device performance. In addition, our 2D-MAP exhibits ferroelectric properties that can facilitate charge transport across the 2D layer due to barrier-lowering effects upon polarization, which are not found in conventional 2D perovskites.⁷ We also measure the charge mobility of control/C₆₀, 2D-PEA-disordered/C₆₀ and 2D-MAP-ordered/C₆₀ that show rapid electron transfer from the perovskite to the C₆₀ for 2D-MAP-ordered samples. This confirms that the restructured 2D-MAP layer does not impact charge transfer, despite its parallel orientation.

Fig. S22: Time-resolved PL (TRPL) spectra perovskite films.

Fig. 3g: Differential carrier lifetimes extracted from the time-resolved PL spectra.

The above revisions have been integrated in the revised manuscript (page 9).

Comment: 6. I cannot agree with the hypothesis of Figure 3g. Transient absorption (TA) measurements would reflect the sign of the whole film rather than the interface charge transfer. In contrast, I suggest the author measure the TA for the 2D/3D perovskite with different treatments

(materials and methods). Thus, the authors should analyze the charge transfer between different phases based on the PB characteristics.

Response: We thank the Reviewer for giving us an opportunity to double-check the TA data. We understand the concern that TA measurements typically reflect the behaviour of the entire film rather than exclusively the interface charge transfer. We also agree that further TA measurements focusing on the 2D/3D perovskite with different treatments would provide a clearer understanding of the charge transfer dynamics between different phases. In particular, analyzing the photobleaching (PB) characteristics will help us evaluate the charge transfer at the interface. Unfortunately, and despite extensive efforts, we were unable to identify any signal associated with the 2D perovskite in our TA data (**Fig. R1**). We expect that our 2D-MAP forms unevenly, as confirmed by the TEM image and discussed above, resulting in a negligible or very weak transient absorption signal. We also failed to detect any XRD signal associated with the 2D perovskite phase (see added **Fig. S30a**). This distinct absence of 2D indicates limited measurement detection capabilities and/or the presence of a very thin and uneven 2D layer atop the 3D phase. Clear evidence of the 2D-MAP layer can be observed using a surface-sensitive characterization like GIWAXS, as shown in **Fig. 1a**.

Fig. R1: TA spectra of perovskite samples on different 2D conditions. We can only see the signals of 3D perovskite and PbI₂, but not the signals of 2D perovskite.

In response, and to minimise confusion, we have removed the TA related data in the revised manuscript. To clarify the charge transfer analysis in our system, we added the time-resolved photoluminescence (TRPL) data. **Fig. S22** and **Table S2** show the TRPL spectra of perovskite films with C₆₀. The MAP/C₆₀ sample shows a longer lifetime, indicating efficient suppression of the interfacial defects induced by the C₆₀ layer. Moreover, for the charge transfer region, we further analyzed the TRPL data to calculate the differential lifetimes associated with the electron transfer from the perovskite to the C₆₀ for the MAP samples.

Fig. S30a: XRD patterns of the 2D-MAP-disordered and 2D-MAP-ordered-based perovskite films.

Fig. S22: Time-resolved PL (TRPL) spectra perovskite films.

Fig. 3g: Differential carrier lifetimes extracted from the time-resolved PL spectra.

The above discussion and data have now been integrated in the revised manuscript and SI file.

Comment: 7. Energy alignment schemes should be given in the main part. In the SI, I found that the over-reduction of the WFs would be harmful.

Response: We appreciate the Reviewer's suggestions. We agree that incorporating energy alignment schemes into the main manuscript would improve the clarity of our findings. In the revised version, we have relocated these schemes from the supplementary information to the main text (**Fig. 3k**). We recognize the Reviewer's apprehension regarding the work function (WF) decrease in our 2D-MAP compared to the 3D control. Our observations indicate that the WF of the control perovskite film changes in aged samples relative to fresh samples, possibly because of partial degradation. In contrast, the 2D-MAP's WF remains similar for both fresh and aged samples. Therefore, we updated the ultraviolet photoelectron spectroscopy (UPS) and WF data for the control sample using a fresh film, resulting in a WF of approximately -4.73 eV for the fresh control sample. This value aligns with our previous findings (Science 2022, DOI:10.1126/science.abm5784). The valence band maximum (VBM) and conduction band minimum (CBM) of the fresh and aged samples show minimal variation. As a result, WF reduction is now less significant and not detrimental.

Comment: 8. Since QFLS has been given, I encourage the author to present the V_{OC} -loss analysis for the direct comparison between different treatments.

Response: We thank the Reviewer for the suggestion. We added the V_{OC} -loss analysis in the revised manuscript, and the results are consistent with the QFLS enhancement. In response we added **Fig. S21b** in the revised SI file.

Fig. S21b: QFLS and V_{OC} loss analysis.

Comment: 9. Could the post-dripping be used in the large-area solution-processing? It is very important for the application of this method. Otherwise, the results of 1.0 cm² devices are not attractive.

Response: We also acknowledge the Reviewer's input regarding scalability, as reviewer 1 also raised the same question (number 8). Our post-dripping method also offers scalability aspects. To explore this, we tested IPA post-dripping on 2D-MAP passivated films using the blade coating or dipping method, which is scalable for large areas. The results showed that IPA post-dripping on 2D-MAP films using blade coating or the dipping method improved 1-cm² device performance compared to cells without post-dripping. This result proves the method's promise for large-scale fabrication compatibility.

In response we added blade- and dip-coating of IPA on 1-cm² devices in **Fig. S19 and S20**.

Fig. S19: Schematic diagram of blade-coating and dip-coating methods. We employed scalable fabrication techniques to validate the effect of IPA post-dripping. For the blade-coating method, IPA post-dripping was performed by blade-coating on substrates at 25°C at a speed of 1.5 m/min in a glove box. A nitrogen (N_2) flow was applied after the blade to accelerate the evaporation of IPA. For the dip-coating method, IPA post-dripping was carried out by immersing the 3D/2D-MAP-disordered film in a beaker of IPA for approximately 3 seconds, followed by drying the film with N_2 flow to remove residual IPA in the glove box.

Fig. S20: Statistical comparison of photovoltaic parameters (V_{oc} , J_{sc} , FF, and PCE) of 2D-MAP-disordered and 2D-MAP-ordered-based devices by blade-coating and dip-coating of IPA on 3D/2D-MAP films ($1.0\ cm^2$).

The above discussion/data has now been integrated in the revised manuscript and SI file.

=====

Reviewer #3 (Remarks to the Author):

Authors report the use of a meta-amidinopyridine (MAP) ligand through a solvent post-dripping step to produce a highly ordered 2D perovskite phase on the surface of 3D perovskite film. The resulting reconstructed 2D/3D perovskite heterointerface showed reduced energetic disorder and enhanced performance compared to the control and randomly oriented 2D/3D perovskite sample. PSCs made with this ordered 2D-phase perovskite achieved a maximum power conversion efficiency (PCE) of 26.05% (a certified value of 25.44%). During damp heat and outdoor tests were also performed to demonstrate the improve stability. The paper might be accepted after addressing the following issues.

Response: We appreciate the Reviewer’s comments and the overall positive recommendation. Next we address all highlighted issues in turn.

Comment: 1. The q_z peak of 2D for 2D-MAP in GIWAX shifts significantly after IPA dripping. Thus, the statement “are expected to react less with the post-dripping solvent or have stronger binding to the inorganic octahedral perovskite matrices” is not entirely accurate—it’s not about reacting less but rather about reconstruction. Also, in Fig. 1a, the x-axis should be labeled as q_{xy} instead of q_z . Additionally, is there any difference observed in the XRD?

Response: We thank the Reviewer’s for carefully examining our GIWAXS results. We agree with the reviewer that the q_z peak of 2D-MAP after IPA post-dripping shifted due to structural reconstruction after post-dripping. That statement explains why the MAP ligand is not dissolved during the post-dripping, in contrast to the PEA ligand in 2D-PEA, which has almost completely dissolved and removed, as indicated by the 2D-PEA disappearance. To avoid confusion, we revised the sentence and added more discussion accordingly. We also revised the miss-labelled in **Fig. 1a**.

In addition, we could not detect the 2D perovskite signal using conventional XRD after IPA post-dripping (see added **Fig. S30a**). The 2D-MAP-ordered only can be observed using surface-sensitive characterization methods like GIWAXS, as shown in **Fig. 1a**.

Fig. S30a: XRD patterns of the 2D-MAP-disordered and 2D-MAP-ordered-based perovskite films. Revised the discussion:

“MAP ligands, on the other hand, are expected to react less with the post-dripping solvent or have stronger binding to the inorganic octahedral perovskite matrices, as 2D-MAP still appears after post-dripping or washing. In addition, the shifting of the q_z peak of the 2D-MAP-ordered film indicates the structural reconstruction of 2D-MAP, potentially forming a smaller 2D crystal and shorter 2D interlayer.”

Comment: 2. TEM images in Fig. 1b are not very convincing because perovskite materials easily decompose under FIB milling process (unless using cryo-FIB) and the highlighted area might be PbI_2 instead of 2D perovskite.

Response: We agree with the Reviewer's assessment that TEM measurements can lead to sample decomposition during the FIB milling. In our study, we took specific steps to mitigate the beam damage during FIB and TEM.

Conventional TEM and FIB techniques pose significant challenges when examining the cross-sectional structure of perovskite films owing to their susceptibility to degradation. To address this issue, we employed an advanced automatic FIB-SEM Helios G5 instrument (SEM-FIB Helios G5 DualBeam, FEI). The final milling steps for the TEM lamella were performed at ultra-low currents (7 pA-2 keV) for a prolonged time to minimize damage significantly. Additionally, to mitigate damage during HR-STEM, we used the advanced Double Cs corrected Thermo Fisher™ Themis 60-300 Cubed TEM microscope, operating at 80 kV. During TEM experiments, the beam dose is reduced to 2-5 pA by various instrumental means, compared to 0.2-3 nA in conventional 300 kV TEM. We deliberately avoided cryo-FIB because extensive cooling of the FIB holder and grid increases the likelihood of attracting unwanted moisture, which is detrimental to perovskites. Our techniques collectively enhance the accuracy and resolution of STEM imaging while preserving the integrity of the perovskite films. The lattice parameters (c-interplanar distance) of different crystal structures vary, with 2D perovskites ($> 9.1 \text{ \AA}$) having a larger distance compared to PbI_2 ($d = 6.9 \text{ \AA}$). Through a comparison of lattice spacings, we confirmed that the highlighted area corresponds to a 2D perovskite crystal domain rather than PbI_2 species.

Comment: 3. The N 1s spectra in MAP need fitting, as there are two different N bonds (N in FA, MA, and pyridinic nitrogen atoms). Therefore, the statement that N shifts significantly is not strongly supported.

Response: In response, we have re-fitted the N 1s spectra to account for the different nitrogen bonds, specifically N in FA, MA, and pyridinic nitrogen atoms. The positions of these peaks have been clearly identified in the revised fitting. From the results, we can observe the shifts in N associated with FA and MA. We have updated the manuscript to reflect these findings accordingly. Revised **Fig. S5a** in the SI file.

Fig. S5a: XPS spectra (N 1s) obtained from the different perovskite layers.

Comment: 4. Fig. 3g, the authors mention that these are TA results; however, they do not specify the exact wavelength for these dynamics. Additionally, the time scale in Fig. 3g is in ns, while in Table S2, it is in ps. How can this data be fitted in such a manner? Could it be TRPL?

Response: We thank the Reviewer for pointing out this discrepancy. The exact wavelength for the dynamics in **Fig. 3g** is 775 nm. Regarding the time scales, we used ps unit in Table S2, which may confuse the readers, therefore, we have converted all time axis in ns unit.

Also, the reviewer 2 suggests that transient absorption (TA) measurements would reflect the sign of the whole film rather than the interface charge transfer. Analyzing the photobleaching (PB) characteristics will help us evaluate the charge transfer at the interface. However, we did not find any signal of 2D perovskite through TA analysis (**Fig. R1**). We expect that our 2D-MAP forms unevenly, as confirmed by TEM image (**Fig. 1c** and **Fig. S4c**), which can result in a negligible or very weak transient absorption signal. We also failed to detect the 2D perovskite signal via XRD measurements (see added **Fig. S30a**). The absence of the 2D phase in the data obtained via TA and XRD is in line with the existence of a very thin and uneven 2D layer. Hence, the 2D-MAP layer only can be observed using surface sensitive characterization like GIWAXS as shown in **Fig. 1a**.

To clarify this point, we deleted the manuscript's description and relevant TA data. To clarify the charge transfer analysis in our system, we added the time-resolved photoluminescence (TRPL) data. **Fig. S22** and **Table S2** show the TRPL spectra of perovskite films with C₆₀. The MAP/C₆₀ sample exhibits a longer lifetime, indicating a reduced number of interfacial defects due to the C₆₀ layer. However on the charge transfer region, we further analyzed the TRPL data to calculate differential lifetimes, indicating rapid electron transfer from the perovskite to the C₆₀ for MAP samples. Please also see our response to point 6 for Reviewer 2 above.

Fig. R1: TA spectra of perovskite samples on different 2D conditions. We can only see the signals of 3D perovskite and PbI₂, but not the signals of 2D perovskite.

Fig. S30a: XRD patterns of the 2D-MAP-disordered and 2D-MAP-ordered-based perovskite films.

Fig. S22: Time-resolved PL (TRPL) spectra perovskite films.

Fig. 3g: Differential carrier lifetimes extracted from the time-resolved PL spectra.

Comment: 5. The authors mention “reveal that the conduction band minimum (CBM) of 2D-PEA is higher than that of the control, blocking electron transfer from the perovskite to the C_{60} electron selective contact”, while I can’t find the CBM data for 2D-PEA, only for control and 2D-MAP in **Fig. S17**. Also, based on the UPS results, the calculated energy level for control and 2D-MAP is -3.94/-5.5 eV and -3.84/-5.51 eV. Based on this, the 2D-MAP seems to block the electron extraction of control. Can the authors elaborate more?

Response: We thank the Reviewer for raising this point. Using UPS, we obtained the valence band maximum (VBM) for the 3D perovskite (Control), 2D-PEA-disordered, and 2D-MAP-ordered layers. Additionally, using IPES, we obtained the conduction band minimum (CBM) for the 3D perovskite (Control) and 2D-MAP-ordered layers. Due to limited source and time of measurement of IPES, the CBM of 2D-PEA is calculated based on the bandgap of the 2D $(PEA)_2PbI_4$ perovskite

which around 2.33 eV using formula, $E_{\text{VBM}} = E_{\text{CBM}} - E_{\text{bandgap}}$.^{15,16} The summary of the VBM and CBM values of each samples is summarized in added Table S5 in the revised SI file.

We agree that our 2D-MAP still has an energetic offset of about 9 meV compared to the CBM of the 3D control perovskite, while 2D-PEA has a much higher energetic offset of around 70 meV. We expect this 9 meV difference to have minimal impact on charge transport, while the discontinuous 2D distribution may also minimize the electron blocking issue. Please see also our response to Reviewer #2, point 2, above.

Table S4 has now been added in the SI file.

Table S4: The energy level data of control, PEA and MAP-based perovskite films from UPS and IPES. *This CBM value is obtained using formula, $E_{\text{VBM}} = E_{\text{CBM}} - E_{\text{bandgap}}$ with the bandgap of 2D (PEA)₂PbI₄ perovskite is about 2.33 eV.¹⁵

Perovskites	WF (eV)	VBM (eV)	CBM (eV)
3D Control	4.73	5.50	3.93
2D-PEA ($n = 1$)	4.61	5.61	3.23*
2D-MAP ($n = 1$)	4.46	5.51	3.84

Comment: 6. Regarding the use of novel ligand and 2D structures in 2D/3D perovskite solar cells, some recent work on conjugated molecular design (e.g., Science Advances 2023, 9, eadg003) could be cited.

Response: We appreciate the reviewer's suggestion. We cited the recommended paper to our list of references (Ref 16) in the revised manuscript.

Reference

1. Jiang, R. *et al.* Insights into the effects of oriented crystallization on the performance of quasi-two-dimensional perovskite solar cells. *Next Materials* **1**, 100044 (2023).
2. Cao, D. H., Stoumpos, C. C., Farha, O. K., Hupp, J. T. & Kanatzidis, M. G. 2D homologous perovskites as light-absorbing materials for solar cell applications. *J. Am. Chem. Soc.* **137**, 7843-7850 (2015).
3. Wang, J. *et al.* Templated growth of oriented layered hybrid perovskites on 3D-like perovskites. *Nat. Commun.* **11**, 582 (2020).
4. Gong, C. *et al.* Functional-Group-Induced Single Quantum Well Dion-Jacobson 2D Perovskite for Efficient and Stable Inverted Perovskite Solar Cells. *Adv. Mater.* **36**, 2307422 (2024).
5. Azmi, R. *et al.* Double-side 2D/3D heterojunctions for inverted perovskite solar cells. *Nature* **628**, 93-98 (2024).
6. Ugur, E. *et al.* How humidity and light exposure change the photophysics of metal halide perovskite solar cells. *Sol. RRL* **4**, 2000382 (2020).
7. Pica, G. *et al.* Photo-ferroelectric perovskite interfaces for boosting V_{oc} in efficient perovskite solar cells. *Nat. Commun.* **15**, 8753 (2024).
8. Jiang, Q., *et al.* Surface passivation of perovskite film for efficient solar cells. *Nat. Photonics* **13**, 460-466 (2019).
9. Chen, H. *et al.* Quantum-size-tuned heterostructures enable efficient and stable inverted perovskite solar cells. *Nat. Photonics* **16**, 352-358 (2022).
10. Wang, M. *et al.* Ammonium cations with high pKa in perovskite solar cells for improved high-temperature photostability. *Nat. Energy* **8**, 1229-1239 (2023).

11. Wang, Y. *et al.* Homogenized contact in all-perovskite tandems using tailored 2D perovskite. *Nature* (2024).
12. Perini, C. A. R. *et al.* Interface reconstruction from Ruddlesden–Popper structures impacts stability in lead halide perovskite solar cells. *Adv. Mater.* **34**, 2204726 (2022).
13. Sutanto, A. A. *et al.* Dynamical evolution of the 2D/3D interface: a hidden driver behind perovskite solar cell instability. *J. Mater. Chem. A* **8**, 2343-2348 (2020).
14. Park, S. M. *et al.* Engineering ligand reactivity enables high-temperature operation of stable perovskite solar cells. *Science* **381**, 209-215 (2023).
15. Cho, K. T. *et al.* Selective growth of layered perovskites for stable and efficient photovoltaics. *Energy Environ. Sci.* **11**, 952-959 (2018).
16. Azmi, R. *et al.* Damp heat-stable perovskite solar cells with tailored-dimensionality 2D/3D heterojunctions. *Science* **376**, 73-77 (2022).

Reply to Reviewers' Comments

We are pleased to submit the revised version of our manuscript (NCOMMS-24-48452A) for reconsideration in *Nature Communications*. We thank the Reviewers for assessing our work and providing such constructive feedback. Next, we address all points raised in turn. All revisions are highlighted in the revised manuscript files.

Kind regards,

Prof. Thomas Anthopoulos & co-authors

Reviewer #1 (Remarks to the Author):

All the comments have been addressed well, accept.

Response: We sincerely thank the Reviewer for the positive recommendation.

Reviewer #2 (Remarks to the Author):

The authors have responded to the concerns in the revised manuscript. However, some of the critical questions have not been settled.

Response: We appreciate your critical questions. In response, we have attempted to further clarify the outstanding questions by including additional data and relevant discussions. We sincerely hope the corrections will satisfy the Reviewer.

1. For Comment 3, the authors suggested that the post-dripping process helps to remove excess ligands and reorient the 2D perovskite domains from a random orientation to a highly ordered layer. I cannot follow this explanation. IPA treatment would cause some dissolution of the perovskite surface. Besides, why the ordered structure is thermodynamically more stable? And why the 2D film is uneven?

Response:

1.1 Mechanism

In general, the solution post-treatment of 2D ligand leads to excess organic ligand, limiting the 2D orientation arrangement and quality (see Ref. *Nat. Energy* 2024, DOI: 10.1038/s41560-024-01529-3. *Science* 2022, DOI: 10.1126/science.abm5784. *Sol. RRL* 2023, DOI: 10.1002/solr.202201002. *Nat. Commun.* 2023, DOI: 10.1038/s41467-023-43016-5. *J. Am. Chem. Soc.* 2021, DOI: 10.1021/jacs.1c00757. *Adv. Mater.* 2022. DOI: 10.1002/adma.202204726.). Thus, the IPA post-dripping step helps to remove this excess ligand (unbonded) from the surface but will not dissolve the perovskite. Here during the post-dripping step and ligand removal, we found that the IPA post-dripping can regulate the 2D phase orientation from random orientation to highly ordered layer upon multiple post-dripping steps. The combined effect of excess ligand dissolution/removal and the dynamics of the dripping process is believed to drive the observed surface transformation and re-orientation.

We tested several organic solvents commonly used for solvent dripping in perovskite film formation, such as chlorobenzene and ethyl acetate as non-polar solvent and compared them with IPA as polar solvent (see **Fig. S27**). We found that solvents such as IPA that can dissolve the 2D ligands will facilitate the 2D layer reconstruction. The mechanism is involved that IPA can re-dissolve or remove excess ligands; then reorient the 2D perovskite domains from a random orientation to a highly ordered layer. The process involves post-dripping the film with IPA multiple times to ensure the complete orientation arrangement in the in-plane direction, which is thermodynamically preferable (see Ref. *Science* 2022, DOI: 10.1126/science.abm5784. *Nat. Commun.* 2018, DOI: 10.1038/s41467-018-03757-0. *ACS Materials Letters* 2022, DOI: 10.1021/acsmaterialslett.1c00709. *Next Materials* 2023, DOI: 10.1016/j.nxmater.2023.100044. *J. Am. Chem. Soc.* 2015, DOI: 10.1021/jacs.5b03796), as confirmed by newly added GIWAXS data in **Fig. 1a in Revision 1**. This explain that after the 3rd and 4th IPA post-dripping, the 2D layer became

more ordered in the in-plane direction. We also have added a schematic depicting the reorientation process of the 2D-MAP involving ligand dissolution and removal followed by reorientation of 2D-MAP - see revised **Fig. 1d**. These are the critical steps that affect and, ultimately, determine the observed surface restructuring.

Added **Fig. 1a** and **1d** in Revision 1.

Fig. 1a. 2D GIWAXS images of perovskite films with 0.1° X-ray incidence angles.

Fig. 1d. Schematic of reorienting the 2D-MAP perovskite during and after the post-dripping.

The above parts have now been integrated in the revised manuscript.

1.2 Evidence that unreacted MAP ligand is redissolved or removed by IPA post-treatment

To further confirm that IPA effectively removes unreacted 2D ligand molecules from the surface, we qualitatively compared images of perovskite films under different treatment conditions: 3D/2D-MAP films with and without IPA post-dripping (added **Fig. S3a**). We observed that the 3D/2D perovskite film without IPA post-treatment exhibited a noticeable greenish color compared to the control perovskite film (dark color). After 2-4 times of IPA post-treatment, the greenish color gradually faded (becomes darker), and after ten rounds, it almost completely matched the color of the control perovskite film.

As shown in **Fig. 1a,b**, the 2D peak remained strongly detectable after 3 or 4 times washing with IPA, yet the surface color of the 3D/2D perovskite films differed significantly from the control samples. This result suggests that the greenish color could originate from excess MAP ligands on the surface, changing the perovskite film color from dark to greenish color. Notably, after 4th times of IPA post-dripping, unreacted MAP ligands on the perovskite film is almost completely redissolved or removed while at the same time this process can regulate 2D phase layer to be more ordered. To elucidate the 2D proportion with the excess unbonded ligand, we integrated the 2D peaks from the GIWAXS data (Incident angles: 0.5°) for each sample (**Fig. S3b**). We observed that as the number of post-dripping cycles increased from 0 to 4 times, the amount of 2D perovskite continuously increased, suggesting the unreacted

MAP ligand is redissolved in IPA and facilitated in the restructuring of the 2D perovskite phase with preferable parallel orientation. In contrast, when the number of post-dripping cycles reached 10, the amount of 2D perovskite significantly decreased, indicating the “over-washing” of the layer’s surface leads to the dissolution of organic ligands or cations and leaving PbI_2 species on the perovskite surface (see Fig. 1a).

Fig. S3: a, Photos of perovskite films made on ITO glass substrates. b, The integrated area of 2D peak for different IPA post-dripping 3D/2D perovskite films.

In response, we have revised and added the text on page 4-5:

“To confirm the effectiveness of IPA post-dripping for removing unreacted 2D ligands, we prepared 3D/2D-MAP films before and after IPA post-dripping. Before IPA post-dripping, the 3D/2D films appeared greenish, which faded after four IPA post-dripping cycles and became darker in color (Fig. S3a). Interestingly, after 10 cycles, the color of the 3D/2D-MAP film became similar to that of the control film. Thus, we speculate that the greenish color can be associated with the excess unreacted MAP ligand on the surface of the film. To confirm the existence of 2D-MAP and their proportion, we integrated the 2D peaks from the GIWAXS data for each sample (Fig. S3b). We observed that as the number of post-dripping cycles increased from 0 to 4, the amount of 2D perovskite continuously increased, suggesting that the unreacted MAP ligand was redissolved in IPA and facilitated the reconversion to 2D perovskite with preferable parallel orientation.”

Additionally, we added Fig. S3 in SI file.

1.3 Evidence that post-treatment with IPA does not significantly damage perovskite films

We acknowledge the Reviewer’s concern regarding the potential dissolution of the perovskite film due to IPA post-dripping. IPA post-dripping can dissolve the cations (FA^+ or MA^+) on the surface of 3D perovskites and create defects on the 3D perovskite surface. However, during the initial 2D ligand treatment, this process facilitates cation exchange between the 3D perovskite and the 2D ligand (PEA or MAP), forming a PbI_2 template for the 2D perovskite layer. This process in forming 3D/2D bilayer perovskites is been reported and studied previously (*J. Am. Chem. Soc.* 2021, DOI: 10.1021/jacs.1c00757).

For MAP-based 3D/2D films, repeated IPA post-treatment gradually dissolves the unreacted ligand molecules, which are then removed from the perovskite surface along with the IPA. Simultaneously, the dissolved ligands in IPA participate in re-orientating the 2D perovskite structure or contribute further to healing MAI or FAI deficiency on the perovskite surface.

To confirm that our IPA post-dripping did not significantly damage the perovskite film, we conducted photoluminescence (PL) characterizations on pristine 3D perovskite films and those treated with PEA and MAP ligands. **Fig. S4** illustrates the PL spectra of the 3D perovskite and 3D/2D perovskite films before and after four times of IPA post-dripping. The results demonstrate that the PL intensity of the 3D perovskite film decreased after IPA post-dripping. This decrease is attributed to the IPA-induced dissolution of the 3D perovskite, leading to the formation of additional defects (Ref. *J. Am. Chem. Soc.* 2021, DOI: 10.1021/jacs.1c00757). The trend differed for films treated with PEA and MAP. After four times of IPA post-dripping, the PL intensity slightly increases in the case of PEA, which may be due to the superior passivation effect of molecular PEA compared to 2D-PEA passivation, as previously reported (*Nat. Photonics* 2019, DOI: 10.1038/s41566-019-0398-2). Conversely, the 2D-MAP passivated perovskite film exhibits a more substantial increase in PL intensity, indicating an enhanced passivation effect of the highly ordered 2D-MAP passivation layer. These findings suggest that our IPA post-dripping process did not significantly damage the perovskite film of the 3D/2D-MAP sample.

Fig. S4: a-c, Comparison of PL spectra for 3D, 2D-PEA-treated, and 2D-MAP-treated perovskite films.

In response, we have revised the text on page 5:

“Subsequently, we also evaluated the impact of IPA post-dripping on 3D perovskite films quality by analyzing the steady-state PL spectra of untreated 3D films and those treated with PEA and MAP ligands (see **Fig. S4**). The control 3D film exhibited diminished PL intensity owing to IPA-induced dissolution and defect formation, which is consistent with previous findings (Ref. *J. Am. Chem. Soc.* 2021, DOI: 10.1021/jacs.1c00757). In contrast, the 2D-passivated films demonstrated increased PL intensity. The 2D-PEA sample displayed a slight PL enhancement post-dripping, indicating superior molecular PEA passivation compared to 2D passivation (*Nat. Photonics* 2019, DOI: 10.1038/s41566-019-0398-2). The 2D-MAP passivation sample exhibited significant PL enhancement after post-dripping, suggesting superior passivation from the ordered 2D-MAP layer. These results confirm that IPA post-dripping enhances 2D passivation without significantly compromising the 3D/2D-MAP film.”

Additionally, we added **Fig. S4** in SI file.

1.4 Why the ordered structure is thermodynamically more stable?

We thank the Reviewer for the critical question and for giving us the opportunity to revise the discussion accordingly. We made a terminology mistake on naming the thermodynamically more stable parallel-oriented 2D layer. We have revised this as thermodynamically preferable. This is because, in general, 2D perovskite orientation is parallel due to

a more favorable arrangement and lower energy formation (Ref. *Science* 2022, DOI: 10.1126/science.abm5784. *Nat. Commun.* 2018, DOI: 10.1038/s41467-018-03757-0. *ACS Materials Letters* 2022, DOI: 10.1021/acsmaterialslett.1c00709. *Next Materials* 2023, DOI: 10.1016/j.nxmte.2023.100044. *J. Am. Chem. Soc.* 2015, DOI: 10.1021/jacs.5b03796).

During 2D formation, we typically applied a higher temperature to enhance 2D perovskite formation kinetics, leading to a random distribution of orientations. Thus, it is necessary to regulate the 2D perovskite crystal after finishing the formation of the 3D/2D bilayer using one of the techniques that we developed post-dripping. Through multi-step IPA post-treatment, the partial dissolution of organic cations and ligands and reformation of the 2D phase under the action of the solvent slowed the formation process of the 2D perovskite, allowing it to reorganize into a parallel-oriented.

1.5 Why the 2D film is uneven?

The uneven of 2D perovskite distribution is affected by several factors, such as ligand reactivity, ligand size, and formation kinetics of 2D phase. In general, the long alkyl ligand will form more homogenous 2D layer on top of 3D perovskite (*Nat. energy* 2024, DOI: 10.1038/s41560-024-01667-8 and *Science* 2022, DOI: 10.1126/science.abm5784), while the shorter alkyl-amine ligands tend to form uneven 2D layer. (*Matter* 2023, DOI: 10.1016/j.matt.2023.05.028 and *Nat. energy* 2024, DOI: 10.1038/s41560-024-01529-3) So our 2D-PEA and 2D-MAP will be likely to form uneven 2D phase on top of 3D perovskite, consistent with the previous experimental works. As we are sure the referee appreciates, explaining from first principles the formation of these uneven layers is presently impossible.

2. For Comment 4, my question is that PEA and MAP are obviously not comparable in the case of structural characteristics. I suggest re-selecting the reference object.

Response: This study primarily focuses on regulating 2D capping layer of 3D/2D bilayer using post-dripping solvent. So we believe, comparing the 2D-PEA as most widely used 2D perovskite passivation is still valid and representative enough with our 2D-MAP. We observed that the 2D structure formed by PEAI is easily removed during IPA treatments, primarily due to structural differences between PEAI and MAP. This demonstrates that using common 2D ligands to validate the IPA post-treatment method is reasonable. However, it is important to note that the comparison with PEAI is not the core focus of this paper.

Following the Reviewer's recommendation, we introduced two additional ligands named benzamidinium hydrochloride (BZA) and pyridin-3-ylmethanamine hydrochloride (PMA) ligands. Unlike MAP hydrochloride, BZA lacks the electron-donating pyridine nitrogen, which we hypothesize leads to reduced passivation capabilities (**Fig. S31a**). On the other hand, PMA, like MAP, features an electron-donating pyridine nitrogen, but the amidino group is changed to ammonium group like PEA ligand (**Fig. S31b**). We compared the device performance of BZA and PMA ligands before and after IPA post-treatment. **Fig. S31 c-j** presents the device parameters. The results show significant improvements in V_{OC} and FF after IPA post-dripping, consistent with the behavior observed for MAP. However, due to the absence of pyridine nitrogen, the V_{OC} of the BZA-based device remained around 1.15 V, significantly lower than that of the MAP-based device, even after IPA post-treatment. The performance of PMA-based devices was very close to that of MAP-based devices. Comparing ligands with similar structures and sizes demonstrates that MAP achieves better performance. However, post-dripping is necessary to further enhance the performance of 3D/2D bilayer perovskites.

Fig. S31. a,b, Molecular structure of BZACl and PMAcI. c-j, Statistical comparison of photovoltaic parameters (V_{oc} , J_{sc} , FF, and PCE) of BZA and PMA without post-dripping and BZA and PMA 4th post-dripping-based devices.

In response, we have revised the text on page 10:

“Furthermore, our findings indicate that the IPA post-dripping strategy demonstrated comparable efficacy for other ligands containing amidino or pyridine groups (Fig. S31). This observation underscores the significance of incorporating additional functional groups into the ligand design.”

Additionally, we added Fig. S31 in SI file.

3. For Comment 6, if the 2D film is highly ordered, the 2D characteristics should be identified. Moreover, I found the characteristic PB peak of 3D perovskite widens when using 2D-ordered. As we know, a wide FWHM indicates an increased defect density, which doesn't make any sense. Pleasd explain it.

Response: From the GIWAXS data we can safely conclude that IPA post-treatment transforms the orientation of the 2D perovskites from random to a more ordered phase (Fig. 1a). Furthermore, from piezoresponse force microscopy (PFM) measurements, we observed that the 3D/2D perovskite films treated with IPA exhibited more pronounced ferroelectricity, which can only be attributed to the improved orientation of the 2D perovskites (Fig. 2a-d. Fig. S9-S12). To this end, the enhanced photo-ferroelectricity on 2D passivation has also been reported to help minimize V_{oc} -losses (*Nat. Commun.* 2024, DOI: 10.1038/s41467-024-53121-8). Additionally, cross-sectional SKPM data revealed an enhanced internal electric field at the 3D/ C_{60} junction in ordered-2D-MAP passivated devices, indicating the superior quality of the 2D passivation layer. These findings collectively provide evidence that an ordered 2D passivation layer is beneficial by enhancing the passivation effect.

The ability to detect and assess the quality of the 2D perovskite also depends on its quantity/amount present. To minimize the electron-blocking effect of 2D perovskite, its thickness should be as thin as possible. Due to the thinness of the 2D layer, the 2D signal is not easily detectable in either the 2D-MAP-disordered or 2D-MAP-ordered samples through TA analysis.

As for the broadening of the FWHM, it was partly attributed to the inappropriate scaling of the TA color maps on the data, which failed to highlight the details of the 3D perovskite signals. To address this, we revised the data by readjusting the scale of each TA mapping sample to balance the emphasis on details while covering all data. After these adjustments, we observed no significant change in the FWHM (**Fig. R1**). To further assess the FWHM accurately, we calculated the FWHM from the TA spectra at 10.18 ps (around the midpoint of the delay time) for the control, 2D-MAP-disordered, and 2D-MAP-ordered samples. Similarly, for control/C₆₀, 2D-MAP-disordered/C₆₀, and 2D-MAP-ordered/C₆₀, we calculated the FWHM from the TA spectra at 5.06 ps (around the midpoint of the delay time). Consistent with the TA colour maps, the FWHM differences among these samples were negligible (**Fig. R2**).

It is important to note that we did not include TA mapping in our manuscript due to the invisibility of 2D PB on the samples to avoid confusion.

Fig. R1: TA color map of perovskite samples on different 2D conditions.

Fig. R2: a-c, TA spectra at selected timescales and the FWHM.

Reviewer #3 (Remarks to the Author):

My questions have been well-addressed and in my opinion it can be accepted now.

Response: We thank the Reviewer for the positive recommendation.